# Prune Redundancy, Preserve Essence: Vision Token Compression in VLMs via Synergistic Importance-Diversity

**Zhengyao Fang[1]\*, Pengyuan Lyu[3]\*, Chengquan Zhang[3], Guangming Lu[1], Jun Yu[1], Wenjie Pei[1,2]†**

[1]Harbin Institute of Technology, Shenzhen, [2]Peng Cheng Laboratory, [3]Independent Researcher

{zhengyaonineve,wenjiecoder}@outlook.com,{lvpyuan, zchengquan}@gmail.com,
{yujun, luguangm}@hit.edu.cn

## Abstract

Vision-language models (VLMs) face significant computational inefficiencies caused by excessive generation of visual tokens. While prior work shows that a large fraction of visual tokens are redundant, existing compression methods struggle to balance *importance preservation* and *information diversity*. To address this, we propose PRUNESID, a training-free Synergistic Importance-Diversity approach featuring a two-stage pipeline: (1) Principal Semantic Components Analysis (PSCA) for clustering tokens into semantically coherent groups, ensuring comprehensive concept coverage, and (2) Intra-group Non-Maximum Suppression (NMS) for pruning redundant tokens while preserving key representative tokens within each group. Additionally, PRUNESID incorporates an information-aware dynamic compression ratio mechanism that optimizes token compression rates based on image complexity, enabling more effective average information preservation across diverse scenes. Extensive experiments demonstrate state-of-the-art performance, achieving **96.3%** accuracy on LLaVA-1.5 with only **11.1%** token retention, and **92.8%** accuracy at extreme compression rates (**5.6%**) on LLaVA-NeXT, outperforming prior methods by **2.5%** with **7.8×** **faster** prefilling speed compared to the original model. Our framework generalizes across diverse VLMs and both image and video modalities, showcasing strong cross-modal versatility. Code is available at https://github.com/ZhengyaoFang/PruneSID

## 1 Introduction

Building upon the success of large language models (LLMs) Brown et al. (2020); Achiam et al. (2023); Touvron et al. (2023), vision-language models (VLMs) Bai et al. (2023); Wang et al. (2024); Wu et al. (2024); Yao et al. (2024); Li et al. (2024) have emerged as a powerful paradigm for multimodal reasoning by encoding images into sequences of visual tokens, thereby enabling joint linguistic and visual understanding. However, this approach introduces substantial computational inefficiencies: contemporary VLMs such as LLaVA-1.5 Liu et al. (2023) and LLaVA-NeXT Liu et al. (2024b) typically generate 576 and 2880 visual tokens per image, far exceeding what is necessary to capture the essential semantic content of the image. While empirical study Chen et al. (2024a) demonstrates that approximately 70% of visual tokens can be discarded with negligible accuracy degradation, existing compression methodologies fail to optimally reconcile the dual objectives of *importance-aware selection* and *information diversity* at high compression ratios, significantly limiting their practical utility for general-purpose VLM applications.

Current visual token reduction techniques can be broadly classified into two paradigms, each exhibiting distinct limitations: Attention-guided selection methods retain visual tokens based on their attention scores Arif et al. (2025); Yang et al. (2024); Zhang et al. (2024c). While effective at preserving semantically salient regions, these approaches systematically neglect contextual background information, thereby compromising scene comprehension. This paradigm suffers from two critical shortcomings: (i) redundant token retention, wherein multiple high-attention patches capture visually similar object segments, inefficiently allocating model capacity to duplicated content; and (ii)

---

*Equal contribution.

†Corresponding author.

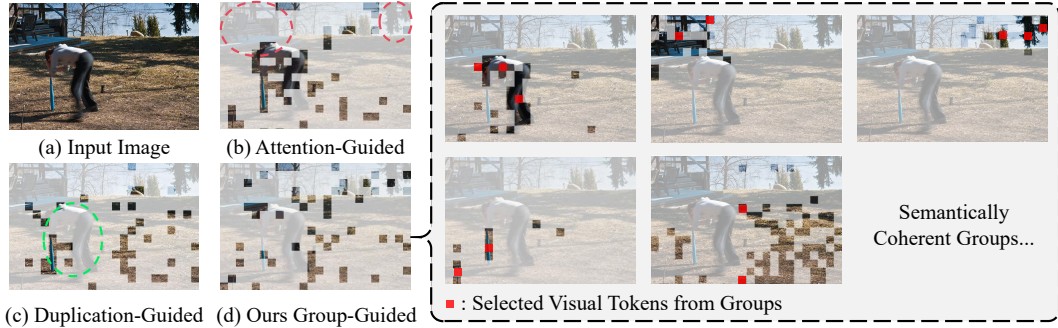

(a) Input Image  (b) Attention-Guided

(c) Duplication-Guided  (d) Ours Group-Guided

Semantically
Coherent Groups...

■ : Selected Visual Tokens from Groups

Figure 1: Comparison of visual token reduction paradigms in VLMs. (a) Original input image. (b) Attention-guided methods preserve high-attention tokens but discard contextual background. (c) Duplication-aware methods remove redundant tokens via similarity pruning, yet may discard semantically important regions with high attention. (d) Our proposed semantically group-guided method balances semantic importance and information diversity.

contextual degradation, as illustrated in Fig. 1 (b), where the lack of attention to background regions leads to incomplete scene information and weaker overall understanding.

Duplication-aware approaches, exemplified by DART Wen et al. (2025) and DivPrune Alvar et al. (2025), address redundancy through similarity-based pruning. However, these methods exhibit a fundamental limitation: the pruning process inadequately considers token-level semantic importance. Consequently, they may fail to retain tokens with high attention scores that are semantically critical, potentially resulting in incomplete or distorted feature representations, as shown in Fig. 1 (c). These observations reveal an inherent trade-off in token compression: *attention-guided methods preserve local salience at the expense of information diversity, while duplication-aware approaches improve diversity while sacrificing salience preservation.*

To address these limitations, we introduce group-guided PRUNESID, an efficient, generic, and training-free framework that achieves task-agnostic token compression while simultaneously optimizing for both importance preservation and information diversity. Our solution employs a novel two-stage pipeline: (1) Principal Semantic Components Analysis (PSCA), which leverages PCA-driven decomposition Abdi & Williams (2010) to automatically cluster tokens into multiple semantically coherent groups, ensuring comprehensive coverage of critical visual concepts; and (2) Intra-group Non-Maximum

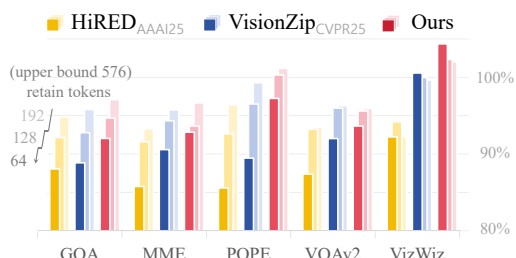

Figure 2: Performance comparison of token reduction methods across multiple vision-language benchmarks on LLaVA-1.5.

Suppression (NMS), which adaptively prunes redundant tokens within each group using dynamic pairwise similarity thresholds (inspired by object detection NMS Neubeck & Van Gool (2006)) while preserving the most semantically significant representatives, as illustrated in Fig. 1 (d). This dual-stage mechanism fundamentally resolves the core trade-off between concept coverage and information density that plagues existing approaches.

Furthermore, PRUNESID incorporates an information-aware dynamic compression ratio mechanism that optimally distributes the token budget per image based on content complexity. This innovation addresses a key limitation of static compression methods by automatically adapting to varying visual semantics, from dense, cluttered scenes to sparse, uniform backgrounds. Our method computes an image-level information score from global token similarity distributions, allocating more tokens to semantically rich images while applying stronger compression to simpler ones. Crucially, this adaptive strategy significantly enhances average information preservation for datasets with high inter-image variability, thereby improving overall model performance.

As demonstrated in Fig. 2, PRUNESID establishes new state-of-the-art performance across multiple vision-language architectures and tasks. The framework achieves 96.3% accuracy on LLaVA-1.5 Liu et al. (2023) while using only 64 tokens (11.1% retention), surpassing VisionZip (92.5%) and HiRED

(87.9%) by significant margins. Remarkably, at extreme compression rates (5.6% tokens), it maintains 92.8% accuracy on LLaVA-NeXT Liu et al. (2024b), representing a 2.5 percentage point improvement over prior approaches. Furthermore, PRUNESID demonstrates exceptional scalability by achieving new SOTA results on Video-LLaVA Lin et al. (2023) with merely 6.6% token retention, confirming its efficacy for both image and video modalities.

In summary, our work makes three principal contributions:

- We propose a training-free visual token compression framework in VLMs that resolves the importance–diversity trade-off via a two-stage pipeline: PSCA for semantic clustering and intra-group NMS for redundancy pruning.
- We introduce an information-aware dynamic compression ratio mechanism that computes a global image-level information score to dynamically assign token budgets across images, enabling effective information preservation in both cluttered and simple scenes.
- Extensive experiments show that our method outperforms prior state-of-the-art across multiple VLMs and tasks, achieving up to 2.5% accuracy gains at extreme compression rates (e.g., 5.6% retention), with strong generalization to image and video modalities.

## 2 VISUAL TOKEN REDUCTION IN VLMS

Recent works have identified significant redundancy among visual tokens in VLMs, motivating a line of research focused on training-free methods to improve inference efficiency. A group of approaches Zhang et al. (2024c); Chen et al. (2024b); Wen et al. (2025); Liu et al. (2024c); Dhouib et al. (2025); Yang et al. (2025); Xing et al. (2024); Pei et al. (2025); Li et al. (2025) conducts token pruning in the layers of LLMs. For example, SparseVLM Zhang et al. (2024c) retains visual tokens that receive high average attention scores from textual tokens, indicating stronger textual relevance. Similarly, FastV Chen et al. (2024b) keeps tokens that receive high attention from other tokens, assuming they carry critical information. DART Wen et al. (2025) computes pairwise similarities among tokens and prunes highly similar ones, aiming to retain a less redundant token set. While effective to some extent, these methods still require full token processing in the early LLM layers, incurring non-negligible computational overhead.

To further enhance efficiency, some methods Shang et al. (2024); Arif et al. (2025); Yang et al. (2024); Zhang et al. (2025b;a; 2024b); Zou et al. (2025) apply token compression in the vision encoder stage, performing *early compression* before interfacing with the LLM. LLaVa-PruMerge Shang et al. (2024) proposes an adaptive selection strategy that leverages the sparsity of attention between the CLS token and visual tokens. It selects tokens with high attention, clusters them based on key similarity, and merges them to enhance information density. HiRED Arif et al. (2025) introduces a hierarchical strategy that partitions the image and allocates a token budget to each region based on CLS attention, enabling a more spatially balanced selection of informative tokens. VisionZip Yang et al. (2024) first identifies dominant tokens with strong attention signals and further merges them based on similarity, ensuring the retention of both salient and contextually rich tokens. These methods significantly reduce the input size to the LLM while maintaining competitive performance.

## 3 OUR METHOD

### 3.1 OVERVIEW

Given an input image, a pre-trained vision encoder in VLMs first generates a sequence of visual token embeddings, denoted as $\mathbf{X} = \{\mathbf{x}_1, \ldots, \mathbf{x}_T\} \in \mathbb{R}^{T \times D}$, where $T$ represents the number of tokens and $D$ denotes the embedding dimension. We aim to reduce this token sequence to a compact representation $\widetilde{\mathbf{X}} \in \mathbb{R}^{N \times D}$ with $N \ll T$, while ensuring: (1) maximal preservation of semantically salient visual patterns and (2) maintaining near-complete information integrity for downstream language modeling tasks.

As illustrated in Fig. 3, we present a novel training-free framework for visual token compression in VLMs. Our methodology employs a two-stage processing pipeline: (1) *semantic-aware token grouping via Principal Semantic Component Analysis (PSCA)*, followed by (2) *intra-group redundancy elimination through Non-Maximum Suppression (NMS)*. The PSCA mechanism clusters tokens

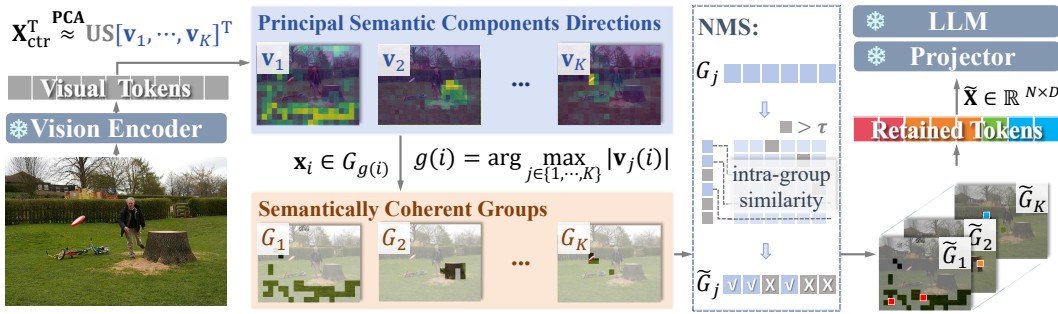

Figure 3: **Overview of our two-stage compression framework.** PSCA first clusters visual tokens into semantically coherent groups via low-rank PCA decomposition. Then, intra-group NMS removes redundant tokens within each group using adaptive similarity thresholds $\tau$, retaining the most informative representatives.

by their contribution to semantic principal component directions, generating groups that maintain both semantic coherence and structural diversity. When integrated with adaptive intra-group pruning, this architecture retains compact yet expressive token representation sets that effectively balance information preservation and diversity.

## 3.2 SEMANTIC-AWARE TOKEN GROUPING VIA PSCA

Unlike conventional PCA Abdi & Williams (2010), which operates in the feature dimension to capture variance-driven directions, Principal Semantic Components Analysis (PSCA) redefines the decomposition objective: it models the *token dimension itself* as the semantic axis of interest. By analyzing cross-token variation, PSCA identifies global semantic directions that reflect coherent visual concepts rather than raw statistical variance. This reframing allows PSCA to uncover latent conceptual structures embedded such as objects, backgrounds, or texture patterns in the token space.

Specifically, given the token embedding matrix $\mathbf{X}$, we first rescale each element via a sigmoid activation $\sigma$ to ensure bounded and comparable feature scales. We then center the features across the token dimension to remove global bias. The resulting mean-centered feature matrix is defined as:

$$\mathbf{X}_{\text{ctr}} = \sigma(\mathbf{X}) - \mu, \quad \text{where} \quad \mu = \frac{1}{T} \sum_{i=1}^{T} \sigma(\mathbf{x}_i) \tag{1}$$

so that $\mathbf{X}_{\text{ctr}} \in \mathbb{R}^{T \times D}$ is the zero-mean token matrix, where each row corresponds to one token. We then apply low-rank PCA decomposition to its transpose matrix $\mathbf{X}_{\text{ctr}}^{\top}$:

$$\mathbf{X}_{\text{ctr}}^{\top} \approx \mathbf{U}\mathbf{S}\mathbf{V}^{\top}, \tag{2}$$

where $\mathbf{V} \in \mathbb{R}^{T \times K}$ contains the top-$K$ right singular vectors that define an orthonormal basis over the token dimension. The columns of $\mathbf{V}$ as $\{\mathbf{v}_1, \ldots, \mathbf{v}_K\}$ represent the principal directions of the components. Each row $|\mathbf{V}_{i,:}|$ indicates how much the $i$-th token contributes to each of the $K$ principal components. A larger value means the token is more strongly related to that component's direction. To form discrete token groups, we assign each token $\mathbf{x}_i$ to the principal direction with the largest absolute value:

$$g(i) = \arg\max_{j} |\mathbf{V}_{i,j}|. \tag{3}$$

This procedure partitions the original $T$ tokens into $K$ semantically coherent groups $\{G_1, \ldots, G_K\}$, each capturing shared semantic information across the image.

## 3.3 INTRA-GROUP REDUNDANCY REMOVAL VIA NMS

The tokens within each group frequently exhibit significant spatial or semantic overlap, particularly in regions containing dense textures or salient objects. To mitigate this redundancy, we employ a non-maximum suppression (NMS) strategy for each group $G_k$, which selectively preserves the most informative tokens while eliminating those demonstrating spatial or semantic redundancy.

Following Eq. 3, each token $\mathbf{x}_i \in \mathbf{X}$ is assigned a selection score $s_i = |V_{i,g(i)}|$, which quantifies its contribution to the principal direction of its assigned group $G_{g(i)}$. We then implement greedy NMS within each group as follows: (1) tokens are ranked by their $s_i$ values, and (2) a token is preserved only if its maximum similarity to all previously selected tokens in $G_k$ falls below a threshold $\tau$. This process generates a refined subset $\widetilde{G}_k \subseteq G_k$ that effectively eliminates redundant tokens while maintaining the original semantic diversity of the group.

To adaptively tune the suppression threshold to varying levels of global redundancy, we introduce a redundancy score $\rho$ defined as the average pairwise similarity among all tokens in the image:

$$\rho = \frac{2}{T(T-1)} \sum_{i=1}^{T} \sum_{j=i+1}^{T} \text{sim}(\mathbf{x}_i, \mathbf{x}_j) \tag{4}$$

where the similarity is computed between $\ell_2$-normalized tokens:

$$\text{sim}(\mathbf{x}_i, \mathbf{x}_j) = \frac{\mathbf{x}_i^\top \mathbf{x}_j}{\|\mathbf{x}_i\| \cdot \|\mathbf{x}_j\|}, \quad \forall \mathbf{x}_i, \mathbf{x}_j \in \mathbf{X}. \tag{5}$$

We then set the NMS threshold as $\tau = \lambda \cdot \rho$, where $\lambda$ is a scaling factor determined by the global token budget $N$. In our experiments, we empirically set $\lambda = \frac{N}{32}$, which consistently worked well across different compression settings. This adaptive threshold encourages stronger suppression for more redundant images.

After performing NMS for all groups, we obtain a collection of filtered groups $\{\widetilde{G}_1, \ldots, \widetilde{G}_K\}$. To match a global token budget $N$, we allocate group-wise quotas $\{n_1, \ldots, n_K\}$ such that $\sum_{k=1}^{K} n_k = N$, where $n_k$ is calculated by rounding $\frac{|\widetilde{G}_k|}{\sum_j |\widetilde{G}_j|} \cdot N$ to the nearest integer.

Finally, we take the top-$n_k$ tokens from each $\widetilde{G}_k$ according to their selection scores $s_i$ and concatenate them to form the final compact token set:

$$\widetilde{\mathbf{X}} = \bigcup_{k=1}^{K} \text{Top}_{n_k}(\widetilde{G}_k). \tag{6}$$

### 3.4 Information-Aware Dynamic Compression Ratio Across Images

Conventional token compression methods employ a fixed token compression ratio $r = \frac{N}{T}$ for all images. This uniform approach leads to suboptimal compression: for complex scenes, the predetermined N proves insufficient, causing excessive information loss; whereas for simple scenes, the same N becomes unnecessarily large, resulting in substantial redundancy.

To address this limitation, we propose an *information-aware dynamic compression ratio strategy* that automatically adjusts the retained token budget $N$ according to each image's information content. Building upon the global redundancy measure $\rho$ from Eq. 4, we compute an image information score:

$$\phi = 1 - \rho, \tag{7}$$

where higher $\phi$ indicates greater semantic diversity and less redundancy. We then allocate the retained token count $N'$ for each image in proportion to its information score: $N' =\propto \phi$. This ensures that more informative images are allocated more tokens, while simpler images are compressed more aggressively, thereby improving compression adaptiveness across diverse scenes.

### 3.5 Theoretical Overview of PSCA–NMS Pruning

Our pruning objective is to retain a token subset $S'$ of fixed size $N$ that maximizes the effective information necessary for VLM. Formally, for a token set $S$, its informativeness is defined as:

$$\text{Inform}(S) = \bigcup_{s_i \in S} I(s_i), \tag{8}$$

where $I(s_i)$ denotes the semantic information of token $s_i$. Using the Inclusion–Exclusion Principle, the informativeness of $S'$ admits the lower bound:

$$\text{Inform}(S') \geq \sum_{s_i \in S'} I(s_i) - \sum_{s_i, s_j \in S'} R(s_i, s_j), \tag{9}$$

where $R(s_i, s_j)$ measures semantic redundancy. PSCA optimizes the first term by selecting tokens with the largest projections onto the principal semantic directions, effectively maximizing $\sum_{s_i \in S'} I(s_i)$. NMS complements this by enforcing a similarity constraint $R(s_i, s_j) \leq \epsilon$, which minimizes redundancy in the second term and ensures diverse semantic coverage. Together, PSCA and NMS jointly approximate the maximization objective:

$$\max_{|S'|=N} \sum_{s_i \in S'} I(s_i) \quad \text{s.t.} \quad R(s_i, s_j) \leq \epsilon, \tag{10}$$

providing a theoretically grounded and effective pruning strategy. The full derivation is presented in Appendix A.1.1.

## 4 EXPERIMENTS

Following the experimental protocol of Yang et al. (2024), we assess the effectiveness of our approach on LLaVA-1.5 Liu et al. (2023). To evaluate generalization, we extend our study to high-resolution vision-language models, including LLaVA-NeXT Liu et al. (2024b) and Mini-Gemini Li et al. (2024). We also conduct experiments on Qwen-VL Wang et al. (2024) in the supplementary material. Evaluations are conducted using LMMs-Eval Zhang et al. (2024a) on a comprehensive suite of widely used visual understanding benchmarks, including GQA Hudson & Manning (2019), MMBench Liu et al. (2024d), MME Fu et al. (2023), POPE Li et al. (2023b), ScienceQA Lu et al. (2022), VQA-v2 Goyal et al. (2017), TextVQA Singh et al. (2019), MMMU Yue et al. (2024), SEED-Bench Li et al. (2023a), VizWiz Gurari et al. (2018), and LLaVA-Bench Liu et al. (2024a). We further evaluate the applicability of our method to video understanding tasks using Video-LLaVA Lin et al. (2023).

### 4.1 MAIN RESULTS ON IMAGE UNDERSTANDING TASKS

**Results on LLaVA-1.5.** LLaVA-1.5 uniformly resizes input images to a resolution of 336×336 before passing them through a CLIP-based Radford et al. (2021) vision encoder, which produces 576 visual tokens. Following prior work Chen et al. (2024b); Zhang et al. (2024c); Yang et al. (2024), we conduct experiments under three token retention settings: 64, 128, and 192 tokens. As shown in Tab. 1, our method consistently achieves state-of-the-art average performance across all configurations, outperforming both early-stage compression approaches that apply compression during the image encoder stage and more computationally intensive methods applied during the prefilling stage. Notably, when retaining only 64 image tokens, our method achieves an average accuracy of approximately **96%** across all benchmarks, surpassing the strong prior method VisionZip by a margin of **1.9%**. This result highlights the superior information richness of the visual tokens selected by our method under extreme compression settings.

PRUNESID-Dyn and VisionZip-Dyn denote the variant of our method and VisionZip augmented with the information-aware dynamic compression ratio mechanism (Sec. 3.4). To ensure a fair comparison, we constrain the average number of retained tokens per benchmark to match that of the fixed-budget setting. Experimental results show that the dynamic strategy consistently improves performance. Notably, its effectiveness varies across benchmarks, which we further analyze in detail in Sec. 4.3.

**Results on LLaVA-NeXT.** LLaVA-NeXT Liu et al. (2024b) divides the image into multiple parts based on its aspect ratio for vision encoding, resulting in a maximum sequence length of up to 2880 tokens. (i.e., 576 tokens × 5). Following the evaluation protocol in Yang et al. (2024), we assess our method under three token retention ratios: 22.2%, 11.1%, and 5.6% of the total visual tokens. The results are presented in Tab. 2. Compared to the strong prior method VisionZip, our approach achieves average performance gains of 0.9%, 1.5%, and **2.5%** under the above three token retention settings, respectively. Notably, even when retaining only about 5% of the original image tokens, our method enables the vision-language model to preserve **92.8%** of its full performance, demonstrating

Table 1: **Performance of PRUNESID on LLaVA-1.5.** *Vanilla* refers to the uncompressed baseline model using all 576 visual tokens, serving as the upper performance bound. *Early Cmp.* indicates whether the compression is applied prior to the LLM for improved efficiency. PRUNESID-Dyn denotes the variant of our method augmented with the Dynamic Compression Ratio mechanism.

| Method | Early Cmp. | GQA | MMB | MME | POPE | SQA | VQA$^{v2}$ | VQA$^{Text}$ | MMMU | SEED | VizWiz | LLaVa-B | Avg. |
|---|---|---|---|---|---|---|---|---|---|---|---|---|---|
| *Upper Bound, 576 Tokens (**100%**)* | | | | | | | | | | | | | |
| Vanilla $_{CVPR24}$ | − | 61.9 | 64.7 | 1862 | 85.9 | 69.5 | 78.5 | 58.2 | 36.3 | 60.5 | 54.3 | 66.8 | 100% |
| *Retain 192 Tokens (↓ 66.7%)* | | | | | | | | | | | | | |
| FastV $_{ECCV24}$ | × | 52.7 | 61.2 | 1612 | 64.8 | 67.3 | 67.1 | 52.5 | 34.3 | 57.1 | 50.8 | 49.4 | 88.2% |
| SparseVLM $_{24.10}$ | × | 57.6 | 62.5 | 1721 | 83.6 | 69.1 | 75.6 | 56.1 | 33.8 | 55.8 | 50.5 | 66.1 | 95.3% |
| MustDrop $_{24.11}$ | × | 58.2 | 62.3 | 1787 | 82.6 | 69.2 | 76.0 | 56.5 | − | − | 51.4 | − | 96.3% |
| DART $_{25.02}$ | × | 60.0 | 63.6 | 1856 | 82.8 | 69.2 | 76.7 | 57.4 | 36.4 | 51.5 | 54.9 | 64.2 | 97.3% |
| ToMe $_{ICLR23}$ | ✓ | 54.3 | 60.5 | 1563 | 72.4 | 65.2 | 68.0 | 52.1 | − | − | − | − | 88.5% |
| LLaVa-PruMerge $_{24.05}$ | ✓ | 54.3 | 59.6 | 1632 | 71.3 | 67.9 | 70.6 | 54.3 | − | − | 50.1 | − | 90.5% |
| FasterVLM $_{2412}$ | ✓ | 59.3 | 63.5 | 1780 | 85.3 | 70.0 | 75.2 | 57.3 | 36.0 | 58.9 | 54.1 | − | 97.9% |
| VisPruner $_{ICCV25}$ | ✓ | 59.4 | 63.3 | 1817 | 85.8 | 70.1 | 75.2 | 57.2 | − | − | 54.3 | − | 98.3% |
| HiRED $_{AAAI25}$ | ✓ | 58.7 | 62.8 | 1737 | 82.8 | 68.4 | 74.9 | 47.4 | − | − | 50.1 | − | 93.6% |
| DivPrune $_{CVPR25}$ | ✓ | 60.0 | 62.3 | 1752 | 87.0 | 68.7 | 75.5 | 56.4 | 35.8 | 58.6 | 55.6 | 64.8 | 96.9% |
| VisionZip $_{CVPR25}$ | ✓ | 59.3 | 63.0 | 1783 | 85.3 | 68.9 | 76.8 | 57.3 | 36.3 | 58.5 | 54.1 | 67.7 | 98.4% |
| VisionZip -Dyn | ✓ | 59.4 | 63.3 | 1797 | 85.5 | 68.7 | 76.9 | 57.4 | 36.8 | 58.6 | 54.4 | 67.7 | 98.6% |
| PRUNESID | ✓ | 60.1 | 63.7 | 1791 | 86.9 | 68.5 | 76.8 | 56.7 | 36.1 | 59.0 | 55.4 | 65.1 | **98.5%** |
| PRUNESID-Dyn | ✓ | 60.2 | 63.8 | 1797 | 87.1 | 69.1 | 76.8 | 56.9 | 36.8 | 59.0 | 55.5 | 65.1 | **98.6%** |
| *Retain 128 Tokens (↓ 77.8%)* | | | | | | | | | | | | | |
| FastV $_{ECCV24}$ | × | 49.6 | 56.1 | 1490 | 59.6 | 60.2 | 61.8 | 50.6 | 34.9 | 55.9 | 51.3 | 52.0 | 84.5% |
| SparseVLM $_{24.10}$ | × | 56.0 | 60.0 | 1696 | 80.5 | 67.1 | 73.8 | 54.9 | 33.8 | 53.4 | 51.4 | 62.7 | 93.0% |
| MustDrop $_{24.11}$ | × | 56.9 | 61.1 | 1745 | 78.7 | 68.5 | 74.6 | 56.3 | − | − | 52.1 | − | 94.7% |
| DART $_{25.02}$ | × | 58.7 | 63.2 | 1840 | 80.1 | 69.1 | 75.9 | 56.4 | 36.2 | 50.5 | 55.3 | 62.4 | 96.0% |
| ToMe $_{ICLR23}$ | ✓ | 52.4 | 53.3 | 1343 | 62.8 | 59.6 | 63.0 | 49.1 | − | − | − | − | 80.4% |
| LLaVa-PruMerge $_{24.05}$ | ✓ | 53.3 | 58.1 | 1554 | 67.2 | 67.1 | 68.8 | 54.3 | − | − | 50.3 | − | 88.9% |
| FasterVLM $_{2412}$ | ✓ | 57.8 | 62.5 | 1762 | 82.8 | 70.0 | 73.9 | 56.3 | 36.9 | 57.6 | 54.3 | − | 97.0% |
| VisPruner $_{ICCV25}$ | ✓ | 58.0 | 61.9 | 1771 | 84.6 | 69.1 | 73.9 | 57.0 | − | − | 52.7 | − | 96.4% |
| HiRED $_{AAAI25}$ | ✓ | 57.2 | 61.5 | 1710 | 79.8 | 68.1 | 73.4 | 46.1 | − | − | 51.3 | − | 92.2% |
| DivPrune $_{CVPR25}$ | ✓ | 59.2 | 62.3 | 1752 | 86.9 | 69.0 | 74.7 | 56.0 | 36.2 | 57.1 | 55.6 | 66.2 | 96.1% |
| VisionZip $_{CVPR25}$ | ✓ | 57.6 | 62.0 | 1762 | 83.2 | 68.9 | 75.6 | 56.8 | 37.9 | 57.1 | 54.5 | 64.8 | 97.2% |
| VisionZip -Dyn | ✓ | 57.6 | 62.2 | 1770 | 83.5 | 68.9 | 75.8 | 56.9 | 37.5 | 57.8 | 54.7 | 65.3 | 97.5% |
| PRUNESID | ✓ | 58.8 | 62.1 | 1749 | 86.5 | 68.8 | 75.3 | 54.7 | 35.8 | 57.8 | 55.8 | 68.8 | **97.6%** |
| PRUNESID-Dyn | ✓ | 58.9 | 62.6 | 1760 | 86.9 | 68.8 | 75.4 | 55.1 | 36.3 | 57.9 | 56.0 | 68.9 | **98.1%** |
| *Retain 64 Tokens (↓ 88.9%)* | | | | | | | | | | | | | |
| FastV $_{ECCV24}$ | × | 46.1 | 48.0 | 1256 | 48.0 | 51.1 | 55.0 | 47.8 | 34.0 | 51.9 | 50.8 | 46.1 | 76.3% |
| SparseVLM $_{24.10}$ | × | 52.7 | 56.2 | 1505 | 75.1 | 62.2 | 68.2 | 51.8 | 32.7 | 51.1 | 53.1 | 57.5 | 87.6% |
| MustDrop $_{24.11}$ | × | 53.1 | 60.0 | 1612 | 68.0 | 63.4 | 69.3 | 54.2 | − | − | 51.2 | − | 88.9% |
| DART $_{25.02}$ | × | 55.9 | 60.6 | 1765 | 73.9 | 69.8 | 72.4 | 54.4 | 35.9 | 47.2 | 55.3 | 59.1 | 92.6% |
| ToMe $_{ICLR23}$ | ✓ | 48.6 | 43.7 | 1138 | 52.5 | 50.0 | 57.1 | 45.3 | − | − | − | − | 70.1% |
| LLaVa-PruMerge $_{24.05}$ | ✓ | 51.9 | 55.3 | 1549 | 65.3 | 68.1 | 67.4 | 54.0 | − | − | 50.1 | − | 87.2% |
| FasterVLM $_{2412}$ | ✓ | 55.0 | 60.6 | 1667 | 76.6 | 70.2 | 70.6 | 55.3 | 35.4 | 54.7 | 55.7 | − | 94.8% |
| VisPruner $_{ICCV25}$ | ✓ | 55.4 | 60.1 | 1691 | 80.4 | 69.1 | 70.9 | 55.8 | − | − | 53.3 | − | 93.8% |
| HiRED $_{AAAI25}$ | ✓ | 54.6 | 60.2 | 1599 | 73.6 | 68.2 | 68.7 | 44.2 | − | − | 50.2 | − | 88.4% |
| DivPrune $_{CVPR25}$ | ✓ | 57.6 | 59.3 | 1638 | 85.6 | 68.3 | 72.9 | 55.5 | 36.3 | 57.5 | 55.5 | 64.0 | 94.6% |
| VisionZip $_{CVPR25}$ | ✓ | 55.1 | 60.1 | 1690 | 77.0 | 69.0 | 72.4 | 55.5 | 36.2 | 54.5 | 54.8 | 62.9 | 94.0% |
| VisionZip -Dyn | ✓ | 55.2 | 60.1 | 1694 | 77.1 | 69.2 | 72.8 | 55.8 | 36.7 | 54.7 | 54.9 | 63.1 | 94.4% |
| PRUNESID | ✓ | 57.1 | 58.8 | 1733 | 83.8 | 67.8 | 73.7 | 54.2 | 37.0 | 56.1 | 56.9 | 65.2 | **95.9%** |
| PRUNESID-Dyn | ✓ | 57.2 | 59.7 | 1734 | 84.1 | 68.1 | 73.8 | 54.2 | 37.2 | 56.2 | 57.0 | 65.8 | **96.3%** |

its ability to maximize information preservation without introducing task-specific biases, such as overemphasizing foreground content at the expense of contextual or background information.

**Results on Mini-Gemini.** Following Yang et al. (2024), we also evaluate the generalizability of our method on the Mini-Gemini model to demonstrate its effectiveness across diverse VLM architectures. Mini-Gemini incorporates a high-resolution vision encoder based on ConvNeXt-L Liu et al. (2022) to extract fine-grained visual features. We apply our token compression method to the final image tokens produced by the vision encoder and evaluate the model's inference performance under various token retention settings across multiple benchmarks. As shown in Tab. 3, our method consistently delivers strong performance, validating its robustness across architectures with different vision backbones.

## 4.2 MAIN RESULTS ON VIDEO UNDERSTANDING TASKS

To further evaluate the effectiveness of our method on video understanding tasks, we apply PRUNESID to Video-LLaVA Lin et al. (2023) and conduct experiments on four video question answering benchmarks: TGIF Jang et al. (2017), MSVD Xu et al. (2017), MSRVTT Xu et al. (2017), and ActivityNet Yu et al. (2019). Each input video consists of 8 frames, with 256 tokens per frame, resulting in 2048 tokens.

Table 2: Performance on LLaVA-NeXT.

| Method | GQA | MMB | MME | POPE | SQA | VQA$^{v2}$ | MMMU | SEED$^I$ | Avg. |
|---|---|---|---|---|---|---|---|---|---|
| *Upper Bound, 2880 Tokens (100%)* | | | | | | | | | |
| Vanilla | 64.2 | 67.9 | 1842 | 86.4 | 70.2 | 80.1 | 35.1 | 70.2 | 100% |
| *Retain 640 Tokens (↓ 77.8%)* | | | | | | | | | |
| VisionZip | 61.3 | 66.3 | 1787 | 86.3 | 68.1 | 79.1 | 34.7 | 66.7 | 97.5% |
| PRUNESID | 61.6 | 64.2 | 1795 | 86.3 | 68.3 | 78.5 | 37.9 | 67.3 | **98.4%** |
| *Retain 320 Tokens (↓ 88.9%)* | | | | | | | | | |
| VisionZip | 59.3 | 63.1 | 1702 | 82.1 | 67.3 | 76.2 | 35.3 | 63.4 | 94.3% |
| PRUNESID | 60.5 | 63.0 | 1754 | 83.1 | 67.3 | 76.6 | 36.4 | 65.0 | **95.8%** |
| *Retain 160 Tokens (↓ 94.4%)* | | | | | | | | | |
| VisionZip | 55.5 | 60.1 | 1630 | 74.8 | 68.3 | 71.4 | 36.1 | 58.3 | 90.3% |
| PRUNESID | 58.9 | 60.8 | 1704 | 76.9 | 67.1 | 73.8 | 36.2 | 62.5 | **92.8%** |

Table 3: Performance on Mini-Gemini.

| Method | GQA | MMB | MME | POPE | SQA | VQA$^{v2}$ | MMMU | SEED$^I$ | Avg. |
|---|---|---|---|---|---|---|---|---|---|
| *Upper Bound, 576 Tokens (100%)* | | | | | | | | | |
| Vanilla | 62.4 | 69.3 | 1841 | 85.8 | 70.7 | 80.4 | 36.1 | 69.7 | 100% |
| *Retain 192 Tokens (↓ 66.7%)* | | | | | | | | | |
| VisionZip | 60.3 | 68.9 | 1846 | 82.3 | 70.1 | 79.1 | 36.1 | 67.5 | 98.3% |
| PRUNESID | 61.2 | 67.2 | 1842 | 84.4 | 71.1 | 79.1 | 36.1 | 67.8 | **98.7%** |
| *Retain 128 Tokens (↓ 77.8%)* | | | | | | | | | |
| VisionZip | 58.7 | 68.1 | 1841 | 78.5 | 70.0 | 77.5 | 34.8 | 65.6 | 96.2% |
| PRUNESID | 60.1 | 66.6 | 1821 | 82.4 | 70.7 | 77.8 | 36.0 | 66.5 | **97.4%** |
| *Retain 64 Tokens (↓ 88.9%)* | | | | | | | | | |
| VisionZip | 55.8 | 65.9 | 1737 | 69.6 | 70.7 | 73.9 | 35.6 | 61.7 | 92.4% |
| PRUNESID | 58.3 | 63.1 | 1735 | 76.0 | 70.6 | 75.2 | 37.2 | 63.6 | **94.4%** |

Following prior works Chen et al. (2024b); Zhang et al. (2024c); Yang et al. (2024), we compress 256 tokens of each frame into 17 tokens, retaining only 6.6%. This reduces the full 2048 video tokens to just 136, which are then passed to the subsequent stages. As shown in Tab. 4, our method achieves consistently better performance than strong prior approaches across all benchmarks, reaching an average accuracy of **95.5%**. These results highlight the

Table 4: Performance on Video-LLaVA.

| Method | TGIF | MSVD | MSRVTT | A-Net | Avg. |
|---|---|---|---|---|---|
| Video-LLaVA | 47.1 | 70.7 | 59.2 | 43.1 | 100% |
| FastV | 23.1 | 38.0 | 19.3 | 30.6 | 52.1% |
| SparseVLM | 44.7 | 68.2 | 31.0 | 42.6 | 86.5% |
| VisionZip | 42.4 | 63.5 | 52.1 | 43.0 | 93.2% |
| PRUNESID | 45.8 | 67.1 | 53.3 | 43.1 | **95.5%** |

strength of our synergistic importance-diversity approach in preserving key semantically representative information under high compression ratios, thereby enabling stronger generalization across diverse video understanding tasks.

## 4.3 ABLATION STUDY

**Ablation on Token Grouping Strategy.** We evaluate the effect of different token grouping strategies used before the intra-group NMS. Specifically, we compute our PSCA-based grouping method with two alternatives: i) a *random grouping* baseline where tokens are shuffled and uniformly partitioned into groups, and ii) a *KMeans-based grouping* Lloyd (1982) applied directly on the token features. As shown in Tab. 5, PSCA consistently outperforms other methods across four benchmarks. This demonstrates the advantage of PSCA in forming semantically coherent token groups by leveraging the local principal subspace structure, leading to more effective redundancy reduction.

Table 5: Ablation study of group method on LLaVA-1.5.

| Method | Retain 64 | | | | Retain 128 | | | | Retain 192 | | | | Avg. |
|---|---|---|---|---|---|---|---|---|---|---|---|---|---|
| | GQA | MME | POPE | SQA | GQA | MME | POPE | SQA | GQA | MME | POPE | SQA | |
| random | 56.2 | 1707 | 79.6 | 67.5 | 57.8 | 1723 | 84.0 | 66.0 | 59.4 | 1743 | 85.6 | 67.5 | 94.8% |
| kmeans | 56.5 | 1630 | 82.8 | 67.8 | 58.7 | 1714 | 86.3 | 67.8 | 60.0 | 1745 | 86.8 | 68.0 | 95.6% |
| PRUNESID | 57.1 | 1733 | 83.8 | 67.8 | 58.8 | 1749 | 86.5 | 68.3 | 60.1 | 1793 | 86.9 | 68.5 | **96.8%** |

**Ablation on Token Group Counts K.** We study how the number of token groups ($K$) affects PRUNESID performance under different total retained token counts ($N \in \{64, 128, 192\}$). For each $N$, we sweep the number of groups $K$ from 8 to 64. As shown in Tab. 6, performance exhibits a bell-shaped trend: too few groups reduce the granularity of redundancy modeling, while too many groups lead to overly small group sizes and unstable pruning. The optimal settings align with moderate values of $K = \frac{N}{4}$. These results validate our heuristic choice of increasing $K$ proportionally with $N$, balancing diversity and intra-group competition.

**Ablation on ViT Layer Features for PSCA Grouping.** We analyze the effect of using features from different ViT layers for PSCA grouping. As Fig. 4 shows, middle-to-late layers (16, 22) yield better results across multiple metrics, indicating more effective semantic clustering. Early layers (0, 2) underperform due to weaker semantic information. Notably, the final output layer (23) shows a slight drop or plateau compared to layer 22, likely because layer 22 features are directly used for LLM training and thus better capture the semantic information needed for token grouping, whereas layer 23's final output is more specialized and less balanced for this purpose. These results validate our choice to extract intermediate-late layer features (e.g., layer 22) for PSCA, striking a balance between semantic richness and balanced coverage.

Table 6: Ablation study of group $K$ on LLaVA-1.5.

| K | Retain 64 | | | | | Retain 128 | | | | | Retain 192 | | | | |
|---|---|---|---|---|---|---|---|---|---|---|---|---|---|---|---|
| | GQA | MME | POPE | SQA | Avg. | GQA | MME | POPE | SQA | Avg. | GQA | MME | POPE | SQA | Avg. |
| 8 | 56.2 | 1684 | 81.4 | 67.1 | 93.1% | 58.2 | 1720 | 85.3 | 68.3 | 96.0% | 59.2 | 1763 | 83.0 | 67.9 | 96.2% |
| 16 | 57.1 | 1733 | 83.8 | 67.8 | **95.1%** | 58.6 | 1692 | 86.4 | 68.2 | 96.1% | 59.2 | 1755 | 85.4 | 68.3 | 96.9% |
| 32 | 57.1 | 1700 | 83.7 | 68.0 | 94.7% | 58.8 | 1749 | 86.5 | 68.3 | **97.0%** | 60.1 | 1763 | 86.8 | 68.4 | 97.9% |
| 48 | 56.9 | 1668 | 83.7 | 67.6 | 94.1% | 58.7 | 1720 | 85.8 | 68.3 | 96.3% | 60.1 | 1793 | 86.9 | 68.5 | **98.3%** |
| 60 | 56.7 | 1657 | 83.6 | 67.6 | 93.8% | 58.6 | 1715 | 85.7 | 68.0 | 96.1% | 60.0 | 1774 | 86.9 | 68.5 | 98.0% |

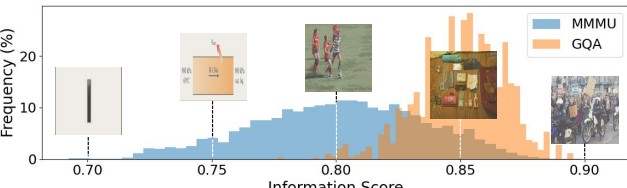

Figure 4: Ablation study on ViT layer features for PSCA Grouping.

**Ablation on Dynamic Compression Ratio Mechanism.** We have demonstrated the effectiveness of the dynamic compression ratio mechanism in Tab. 1. Furthermore, we conduct an in-depth analysis of its performance variations across diverse benchmarks and its generalization capability across multiple model architectures. Theoretically, as the heterogeneity of information scores among test images increases, the adaptive adjustment capacity of the dynamic compression ratio mechanism broadens, thereby amplifying performance enhancements. This hypothesis is corroborated by the distribution depicted in Fig. 5, where the MMMU benchmark demonstrates significantly greater information score variability relative to GQA, a trend consistent with the enhanced performance gains observed for MMMU in Tab. 1.

Figure 5: Histogram of Information Score distributions for the MMMU and GQA benchmarks. A higher Information Score indicates greater visual information content.

To further validate the advantages of the dynamic compression ratio strategy, we conduct comprehensive experiments on LLaVA-1.5, LLaVA-NeXT, and Mini-Gemini across diverse datasets characterized by high information variance, including MME, ScienceQA, MMMU, and POPE. As shown in Tab. 7, our dynamic strategy consistently surpasses fixed-token baselines, achieving up to 1.0 % performance improvements under identical average token budgets. These findings underscore the efficacy of adaptive token compression in average information preserving, particularly beneficial for benchmarks with substantial inter-image variability.

## 4.4 EFFICIENCY ANALYSIS.

Vision-language models (VLMs) suffer from prolonged prefilling time due to excessive visual tokens generated by dense image encoding. As shown in Tab. 8, on the POPE benchmark, LLaVA-NeXT 7B produces up to 2,880 visual tokens per image, where prefilling occupies 86% of the total inference time (254 ms/sample).

For consistency with prior work Yang et al. (2024), we report inference time on a single NVIDIA A800-80GB. At a compression rate of 5.6% (retaining only 160 tokens), our approach reduces prefilling time from 218ms to just 27.8ms, a 7.8× improvement, while also decreasing over-

Table 8: **Efficiency analysis and comparison.** Inference and prefilling times represent the average per-sample latency.

| Method | Token | Inference Time ↓ | Prefilling Time ↓ | POPE (F$_1$)↑ |
|---|---|---|---|---|
| LLaVA-NeXT | 2880 | 254ms | 218ms | 86.4 |
| FastV | 160 | 199ms | 119ms | 50.5% |
| SparseVLM | 160 | 211ms | 128ms | 80.2% |
| VisionZip | 160 | **84ms** | **27.8ms** | 86.6% |
| PRUNESID | 160 | 89ms | **27.8ms** | **89.0%** |

Table 7: Ablation study of dynamic compression ratio mechanism.

| Methods | LLaVA-1.5 | | | | | | LLaVA-NeXT | | | | | | Mini-Gemini | | | | | |
|---|---|---|---|---|---|---|---|---|---|---|---|---|---|---|---|---|---|---|
| | MME | SQA | MMMU | POPE | Avg. | Δ | MME | SQA | MMMU | POPE | Avg. | Δ | MME | SQA | MMMU | POPE | Avg. | Δ |
| baseline | 1862 | 69.5 | 36.3 | 85.9 | 100% | | 1842 | 70.2 | 35.1 | 86.4 | 100% | | 1841 | 70.7 | 36.1 | 85.8 | 100% | |
| | *Retain 192 Tokens (↓66.7%)* | | | | | | *Retain 128 Tokens (↓77.8%)* | | | | | | *Retain 192 Tokens (↓66.7%)* | | | | | |
| PRUNESID | 1791 | 68.5 | 36.1 | 86.9 | 98.8% | | 1795 | 68.3 | 37.9 | 86.3 | 100.7% | | 1842 | 71.1 | 36.1 | 84.4 | 99.7% | |
| PRUNESID-Dyn | 1797 | 69.1 | 36.8 | 87.1 | 99.7% | +0.9 | 1798 | 68.9 | 38.2 | 86.5 | 101.2% | +0.5 | 1845 | 71.3 | 36.9 | 84.6 | 100.5% | +0.8 |
| | *Retain 128 Tokens (↓77.8%)* | | | | | | *Retain 64 Tokens (↓88.9%)* | | | | | | *Retain 128 Tokens (↓77.8%)* | | | | | |
| PRUNESID | 1749 | 68.3 | 35.8 | 86.5 | 97.9% | | 1754 | 67.3 | 36.4 | 83.1 | 97.7% | | 1821 | 70.7 | 36.0 | 82.4 | 98.7% | |
| PRUNESID-Dyn | 1760 | 68.8 | 36.3 | 86.9 | 98.7% | +0.8 | 1787 | 67.7 | 36.8 | 83.5 | 98.7% | +1.0 | 1846 | 71.3 | 36.4 | 82.5 | 99.5% | +0.8 |
| | *Retain 64 Tokens (↓88.9%)* | | | | | | *Retain 32 Tokens (↓94.4%)* | | | | | | *Retain 64 Tokens (↓88.9%)* | | | | | |
| PRUNESID | 1733 | 67.8 | 37.0 | 83.8 | 97.5% | | 1704 | 67.1 | 36.2 | 76.9 | 95.1% | | 1735 | 70.6 | 37.2 | 76.0 | 96.4% | |
| PRUNESID-Dyn | 1734 | 68.1 | 37.2 | 84.1 | 97.9% | +0.4 | 1744 | 67.7 | 36.4 | 77.4 | 96.1% | +1.0 | 1760 | 70.8 | 37.2 | 76.9 | 97.1% | +0.7 |

Figure 6: **Visualization of a limitation of our method.** Left: input image and the corresponding question. Right: tokens retained by our method and the VLM's response after compression. In fine-grained scenarios, our method may discard important local details due to redundancy in the relevant region, leading to incomplete answers.

all inference time to 89ms per sample. Compared to VisionZip, which achieves similar latency (27.8ms prefilling, 84ms inference), our method maintains the same level of efficiency but delivers superior performance on POPE, improving $F_1$ score from 86.6% to 89.0% (+2.4%). This demonstrates that our method not only preserves computational efficiency but also retains more semantically relevant visual information during compression.

### 4.5 VISUALIZATION ABOUT LIMITATION

While the task-agnostic nature of our method provides strong generalization ability, it may be less effective in tasks requiring fine-grained or instruction-specific reasoning. As shown in Fig. 6, when a query focuses on specific visual details, our method may overlook relevant tokens under extreme compression settings (e.g., retaining only 11.1% of tokens). To address this limitation, future work will explore incorporating task-adaptive cues or instruction-aware filtering mechanisms to better align token selection with downstream task requirements.

### 5 CONCLUSION

In this work, we present PRUNESID, a training-free and task-agnostic framework for efficient visual token reduction in vision-language models (VLMs). By integrating Principal Semantic Component Analysis (PSCA) for semantically coherent grouping with intra-group Non-Maximum Suppression (NMS) for redundancy pruning, PRUNESID effectively balances *importance-aware selection* and *information diversity*. Moreover, its dynamic compression ratio mechanism adapts retained token counts based on image complexity, leading to improved overall performance. Extensive experiments demonstrate state-of-the-art results across both image and video VLM benchmarks, retaining ~5% of visual tokens while achieving 92.8% and 95.5% accuracy on LLaVA-NeXT and Video-LLaVA, respectively. These results highlight the potential of semantically group-guided token selection for scaling VLMs to more demanding and resource-constrained settings.

### ACKNOWLEDGEMENTS

This work was supported by the National Natural Science Foundation of China (Grant No. 62372133, 62125201 and U24B20174).

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

# A  APPENDIX

## A.1  ADDITIONAL ALGORITHM AND EFFICIENCY ANALYSIS

### A.1.1  DETAILED THEORETICAL ANALYSIS OF PSCA–NMS PRUNING

**Definition of Information and Redundancy.**  Let Information$(s_i)$, denoted as $I(s_i)$, represent the semantic information carried by token $s_i$. Let Redundancy$(s_i, s_j)$, denoted as $R(s_i, s_j)$, denote the overlapping semantic information shared between tokens $s_i$ and $s_j$. For a token set $S$, the total effective information required for downstream VLM reasoning is defined as:

$$\text{Inform}(S) = \bigcup_{s_i \in S} I(s_i). \tag{11}$$

**Inclusion–Exclusion Principle of Effective Information.**  Given a retained subset $S'$ of fixed size $N$, the informativeness can be expanded by the Inclusion–Exclusion Principle:

$$\text{Inform}(S') = \sum_{s_i \in S'} I(s_i) - \sum_{s_i, s_j \in S'} I(s_i) \cap I(s_j) + \sum_{s_i, s_j, s_k \in S'} I(s_i) \cap I(s_j) \cap I(s_k) - \dots$$
$$\geq \sum_{s_i \in S'} I(s_i) - \sum_{s_i, s_j \in S'} R(s_i, s_j). \tag{12}$$

The first term measures the total semantic contribution of retained tokens, while the second term penalizes redundancy.

**PSCA Optimizes Semantic Importance.**  The first term of Eq. 12, $\sum_{s_i \in S'} I(s_i)$, evaluates how much semantic information is preserved. PSCA provides a principled way to maximize this term: by decomposing the token embedding matrix along the *token dimension*, PSCA identifies the dominant semantic directions in the global embedding space. Tokens with larger projections onto these principal directions possess higher semantic contribution. Thus, ranking tokens by PSCA scores effectively approximates maximizing $I(s_i)$, allowing PSCA to retain globally salient visual semantics while suppressing noisy or low-level details.

**NMS Minimizes Redundancy.**  Even after PSCA, tokens may cluster around spatially adjacent or visually similar regions. This leads to high redundancy, quantified by the second term $\sum_{s_i, s_j \in S'} R(s_i, s_j)$. NMS eliminates tokens whose pairwise similarity exceeds a threshold $\epsilon$:

$$R(s_i, s_j) \leq \epsilon, \quad \forall s_i, s_j \in S'. \tag{13}$$

By bounding semantic overlap among selected tokens, NMS directly minimizes the redundancy term of Eq. 12 and ensures diverse semantic coverage.

**Joint Optimization of Informativeness.**  Combining PSCA and NMS yields the constrained optimization:

$$\text{Prune}_{\text{ours}}(S) = \arg \max_{|S'|=N} \sum_{s_i \in S'} I(s_i) \quad \text{s.t.} \quad R(s_i, s_j) \leq \epsilon, \ \forall s_i, s_j \in G_k. \tag{14}$$

The effective information preserved by our method satisfies the lower bound:

$$\text{Inform}(\text{Prune}_{\text{ours}}(S)) \geq \max_{|S'|=N} \left( \sum_{s_i \in S'} I(s_i) \right) - \sum_k \binom{n_k}{2} \cdot \epsilon. \tag{15}$$

This guarantees that redundancy is upper-bounded by a constant proportional to $\epsilon$, while semantic contribution is maximized via PSCA.

**Summary.**  PSCA enhances *semantic importance* by capturing global principal semantic directions, whereas NMS enforces *diversity* by suppressing redundant tokens. Together, these components jointly approximate the maximization of effective information in Eq. 12, providing a theoretically grounded and efficient pruning strategy.

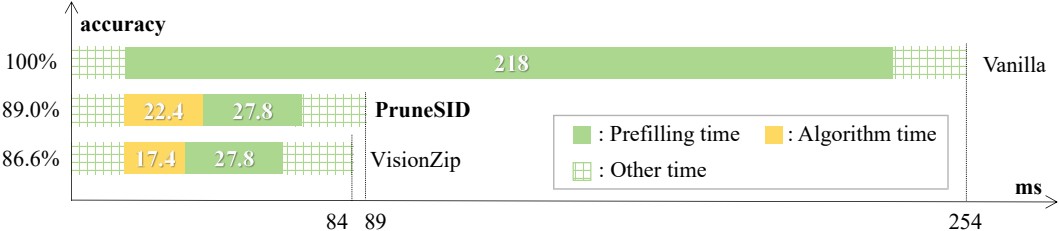

Figure 7: **Runtime comparison of different methods.** *Vanilla* denotes the average runtime of the uncompressed LLaVA-NeXT 7B model on the POPE dataset and the upper performance bound. VisionZip and **PruneSID** represent the runtimes after applying the corresponding compression methods. While maintaining comparable efficiency to VisionZip, our method achieves better performance.

A.1.2    PSEUDOCODE OF OUR METHOD

To provide a clearer understanding of our proposed visual token compression pipeline, we present the full pseudocode of our method. The algorithm consists of two key stages: (1) principal semantic component analysis (PSCA)-based token grouping in Alg. 1, and (2) intra-group non-maximum suppression (NMS) for redundancy removal in Alg. 2. This two-stage design enables both semantic preservation and token diversity under various compression ratios.

---

**Algorithm 1** Semantic-Aware Token Grouping via PSCA

---

**Require:** Visual tokens $\mathbf{X} = [\mathbf{x}_1, \cdots, \mathbf{x}_T] \in \mathbb{R}^{T \times D}$, token budget $N$
1: $\mathbf{V} = \text{pca\_group}(\mathbf{X}, K = \lfloor \frac{N}{4} \rfloor), \mathbf{V} \in \mathbb{R}^{T \times K}$
2: **for** each group $G_k$ **do**
3:     Initialize $G_k \leftarrow \emptyset$
4: **end for**
5: **for** $i = 1$ to $T$ **do**
6:     $g(i) \leftarrow \arg\max_j |\mathbf{V}_{i,j}|$
7:     $G_{g(i)} \leftarrow G_{g(i)} \cup \mathbf{x}_i$
8: **end for**
9: **return** Semantically coherent groups $\{G_1, \cdots, G_K\}$

---

---

**Algorithm 2** Intra-Group Redundancy Removal via NMS

---

**Require:** Groups $\{G_1, \ldots, G_K\}$, visual tokens $\mathbf{X}$, projection matrix $\mathbf{V}$, token budget $N$
1: Compute global redundancy score: $\rho \leftarrow \frac{2}{T(T-1)} \sum_{i<j} \text{sim}(\mathbf{x}_i, \mathbf{x}_j)$
2: Set threshold: $\tau \leftarrow \lambda \cdot \rho$, where $\lambda = \frac{N}{32}$
3: **for** each group $G_k$ **do**
4:     Initialize $\widetilde{G}_k \leftarrow \emptyset$
5:     Sort tokens in $G_k$ by $s_i = |\mathbf{V}_{i,k}|$
6:     **for** each token $\mathbf{x}_i \in G_k$ **do**
7:         **if** $\text{sim}(\mathbf{x}_i, \mathbf{x}_j) < \tau$ for all $\mathbf{x}_j \in \widetilde{G}_k$ **then**
8:             $\widetilde{G}_k \leftarrow \widetilde{G}_k \cup x_i$
9:         **end if**
10:    **end for**
11: **end for**
12: Allocate $n_k \leftarrow \left\lfloor \frac{|\widetilde{G}_k|}{\sum_j |\widetilde{G}_j|} \cdot N \right\rfloor$
13: $\widetilde{\mathbf{X}} \leftarrow \bigcup_k \text{Top}_{n_k}(\widetilde{G}_k)$
14: **return** Final compressed token set $\widetilde{\mathbf{X}}$

---

### A.1.3 RUNTIME COMPARISON AND EFFICIENCY ANALYSIS

While Alg. 1 and 2 are presented with sequential steps for clarity, our actual implementation leverages parallel processing to significantly accelerate execution. The full implementation details are available in our released codebase: https://github.com/ZhengyaoFang/PruneSID.

To further demonstrate the efficiency of our method, we benchmark its runtime under the same hardware and software configuration as used in the main paper's Efficient Analysis section. A detailed comparison with VisionZip Yang et al. (2024) is presented in Fig. 7. On a per-sample basis, our method achieves a processing time of 22.4ms, which is comparable to VisionZip's 17.4ms. Our *early compression* mechanism significantly reduces the total prefilling latency, decreasing it from 218ms to 27.8ms, by reducing the number of visual tokens before they are fed into the LLM.

Despite comparable preprocessing costs, our method consistently yields better performance. Under this configuration, our approach outperforms VisionZip by 2.4% on the POPE dataset, and achieves an average accuracy of 92.8% across multiple benchmarks (see Tab. 2 ↓ 94.4% in the main paper). These results highlight the practicality of our method, offering a favorable trade-off between speed and accuracy.

## A.2 ADDITIONAL EXPERIMENTS

In this section, we present the extended experimental setup and results to validate the effectiveness and generalization ability of our method: (1) we begin by detailing the implementation of our experiments in Sec. A.2.1. (2) To evaluate the generalization of our approach across different model scales, we report results on larger-scale model LLaVA-1.5 Liu et al. (2023) and LLaVA-NeXT Liu et al. (2024b) in Sec. A.2.2. (3) To further assess the applicability of our method to more advanced and architecturally sophisticated vision-language models, we present results on Qwen2-VL Wang et al. (2024) in Sec. A.2.3.

### A.2.1 EXPERIMENT DETAILS

**Environments.** All experiments are conducted using a single NVIDIA L20 GPU with 48GB memory. We build our evaluation pipeline upon the publicly available lmms-eval Zhang et al. (2024a) framework, ensuring consistency with prior benchmarks. For fair comparison in the Efficient Analysis section, we directly benchmark both VisionZip Yang et al. (2024) and our method on the same hardware (L20-48GB) under identical settings. To align with previously reported results obtained using A800-80GB GPUs, we linearly scale our runtime measurements based on VisionZip's performance differential across devices, following standard practice in prior work. This adjustment allows for a meaningful speed comparison while maintaining experimental reproducibility on accessible hardware.

### A.2.2 EVALUATION ON LARGER-SCALE MODELS

In the main paper, we reported the compression results on 7B-scale models. To further evaluate its effectiveness across different model scales, we additionally present results on LLaVA-1.5 13B and LLaVA-NeXT 13B.

LLaVA-1.5 encodes each image into a fixed sequence of 576 visual tokens. Following prior work Yang et al. (2024), we evaluate our method by retaining 192, 128, and 64 tokens, respectively. As shown in Tab. 9, under all three settings, our method achieves average accuracy within 98.0%, 97.4%, and 95.2% of the uncompressed baseline across multiple benchmarks. Compared to the results on 7B-scale models, our method maintains similarly strong compression performance. We further compare our method against VisionZip. Our method outperforms VisionZip by 1% in average accuracy under the most aggressive compression setting of retaining only 64 tokens. When it comes to LLaVA-NeXT 13B in Tab. 10, our method demonstrates competitive performance against VisionZip across different token retention settings, and its advantage becomes particularly prominent under high compression rates. Specifically, under the largest compression rate (94.4% reduction, retaining only 160 tokens), our method's average performance reaches 92.2%, which outperforms VisionZip by 2% (VisionZip: 90.2%). This result fully validates that our method maintains superior visual compression efficiency

Table 9: **Performance of PRUNESID on LLaVA-1.5 13B.** *Vanilla* refers to the uncompressed baseline model using all 576 visual tokens, serving as the upper performance bound.

| Method | GQA | MMB | MME | POPE | SQA | VQA$^{v2}$ | VQA$^{Text}$ | MMMU | SEED$^{I}$ | VizWiz | Avg. |
|---|---|---|---|---|---|---|---|---|---|---|---|
| *Upper Bound, 576 Tokens* (**100%**) | | | | | | | | | | | |
| Vanilla $_{CVPR24}$ | 63.2 | 67.7 | 1818 | 85.9 | 72.8 | 80.0 | 61.3 | 36.4 | 66.9 | 56.6 | 100% |
| *Retain 192 Tokens ( ↓ 66.7%)* | | | | | | | | | | | |
| VisionZip | 59.6 | 65.9 | 1770 | 86.4 | 72.8 | 78.0 | 58.6 | 36.9 | 65.2 | 54.9 | 97.8% |
| PRUNESID | 59.6 | 65.9 | 1770 | 86.4 | 72.8 | 78.0 | 58.6 | 36.9 | 65.2 | 56.0 | **98.0%** |
| *Retain 128 Tokens ( ↓ 77.8%)* | | | | | | | | | | | |
| VisionZip | 57.9 | 66.7 | 1743 | 85.2 | 74.0 | 76.8 | 58.7 | 36.1 | 63.8 | 55.0 | 97.0% |
| PRUNESID | 58.9 | 65.5 | 1811 | 85.9 | 73.1 | 76.7 | 57.5 | 35.8 | 64.1 | 56.8 | **97.4%** |
| *Retain 64 Tokens ( ↓ 88.9%)* | | | | | | | | | | | |
| VisionZip | 56.2 | 64.9 | 1676 | 76.0 | 74.4 | 73.7 | 57.4 | 36.4 | 60.4 | 55.9 | 94.2% |
| PRUNESID | 57.8 | 63.8 | 1711 | 82.0 | 71.8 | 75.2 | 56.3 | 35.7 | 62.8 | 57.3 | **95.2%** |

Table 10: Performance of PRUNESID on LLaVA-NeXT 13B.

| Method | GQA | MMB | MME | POPE | SQA | VQA$^{v2}$ | VQA$^{Text}$ | MMMU | SEED$^{I}$ | VizWiz | Avg. |
|---|---|---|---|---|---|---|---|---|---|---|---|
| *Upper Bound, 2880 Tokens* (**100%**) | | | | | | | | | | | |
| Vanilla $_{CVPR24}$ | 65.4 | 70.0 | 1858 | 86.2 | 73.5 | 81.8 | 64.3 | 36.2 | 71.9 | 64.0 | 100% |
| *Retain 640 Tokens ( ↓ 77.8%)* | | | | | | | | | | | |
| PRUNESID | 62.4 | 67.0 | 1817 | 85.6 | 70.1 | 79.1 | 60.2 | 36.2 | 68.5 | 60.2 | 96.7% |
| *Retain 320 Tokens ( ↓ 88.9%)* | | | | | | | | | | | |
| PRUNESID | 61.5 | 65.4 | 1810 | 82.7 | 70.3 | 76.9 | 58.5 | 36.8 | 66.8 | 58.5 | 95.2% |
| *Retain 160 Tokens ( ↓ 94.4%)* | | | | | | | | | | | |
| PRUNESID | 59.5 | 65.5 | 1715 | 77.8 | 69.1 | 73.6 | 56.7 | 36.4 | 64.1 | 56.7 | 92.2% |

and information preservation capabilities, especially when facing extreme compression demands.

### A.2.3 EVALUATION ON ADVANCED VISION-LANGUAGE ARCHITECTURES

To further assess the generalization of our method on recent, more sophisticated VLMs, we conduct experiments on Qwen-VL Wang et al. (2024), a family of models with notable architectural differences from LLaVA.

Unlike LLaVA's fixed-resolution vision encoders, Qwen-VL employs a dynamically scalable vision encoder that supports arbitrary input resolutions. Moreover, the vision encoder is jointly trained with the language model, enabling tighter cross-modal alignment. After encoding, a lightweight MLP compresses every 2×2 group of neighboring tokens into a single token, resulting in a compact but expressive visual representation.

We evaluate Qwen2-VL 7B under token retention ratios of 33.3%, 22.2%, and 11.1%. As shown in Tab. 11, we first compare our method against PACT Dhouib et al. (2025), the official compression implementation available for Qwen2-VL. Across all three retention levels, our method consistently achieves higher average performance, demonstrating stronger robustness in the low-token regime.

In Tab. 12, we further compare against VisionZip Yang et al. (2024) on Qwen2.5-VL and additionally include two high-resolution benchmarks, HRBench-8k Wang et al. (2025b) and XLRS Wang et al. (2025a). Across all token retention settings, our method consistently outperforms VisionZip, with average performance improvements of +1.6%, +2.2%, and +1.8% at retention ratios of 33.3%, 22.2%, and 11.1%, respectively. Beyond these overall improvements, our method also remains highly effective on the two high-resolution benchmarks. Specifically, on HRBench-8k, our approach preserves 91.6%, 90.7%, and 86.5% of the baseline performance at retention ratios of 33.3%, 22.2%, and 11.1%, respectively. On XLRS, the corresponding preservation ratios are 98.3%, 96.6%, and

Table 11: **Performance of PRUNESID on Qwen2-VL 7B.** *baseline* refers to the uncompressed model using all the visual tokens, serving as the upper performance bound.

| Method | GQA | MMB | MME | POPE | SQA | VQA$^{v2}$ | MMMU | SEED$^I$ | VizWiz | Avg. |
|---|---|---|---|---|---|---|---|---|---|---|
| *Upper Bound* (**100%**) | | | | | | | | | | |
| baseline | 62.3 | 79.1 | 2318 | 87.9 | 84.7 | 81.2 | 51.1 | 76.5 | 68.2 | 100% |
| Token retention ratio 33.3% (↓ **66.7%**) | | | | | | | | | | |
| PACT $_{CVPR25}$ | 59.1 | 75.8 | 2068 | 85.6 | 80.5 | 78.2 | 48.2 | 74.0 | 67.9 | 95.3% |
| PRUNESID | 60.5 | 72.1 | 2146 | 87.0 | 81.8 | 79.8 | 48.3 | 75.5 | 66.1 | **96.1%** |
| Token retention ratio 22.2% (↓ **77.8%**) | | | | | | | | | | |
| PACT $_{CVPR25}$ | 55.8 | 72.2 | 2012 | 82.4 | 78.3 | 77.6 | 48.8 | 71.7 | 67.4 | 92.8% |
| PRUNESID | 59.2 | 69.2 | 2119 | 86.0 | 79.5 | 78.1 | 47.3 | 74.8 | 64.8 | **94.1%** |
| Token retention ratio 11.1% (↓ **88.9%**) | | | | | | | | | | |
| PACT $_{CVPR25}$ | 50.1 | 63.1 | 1785 | 71.4 | 75.0 | 74.3 | 48.5 | 66.0 | 63.1 | 85.8% |
| PRUNESID | 55.9 | 65.0 | 2039 | 82.9 | 76.1 | 73.6 | 46.9 | 71.8 | 64.1 | **90.4%** |

Table 12: **Performance of PRUNESID on Qwen2.5-VL 7B.** *baseline* refers to the uncompressed model using all the visual tokens, serving as the upper performance bound.

| Method | GQA | MMB | MME | POPE | SQA | VQA$^{v2}$ | HRB$^{8k}$ | XLRS$^{macro}$ | Avg. |
|---|---|---|---|---|---|---|---|---|---|
| *Upper Bound* (**100%**) | | | | | | | | | |
| baseline | 60.9 | 83.9 | 2310 | 86.3 | 88.9 | 82.9 | 68.1 | 47.1 | 100% |
| Token retention ratio 33.3% (↓ **66.7%**) | | | | | | | | | |
| VisionZip $_{CVPR25}$ | 56.6 | 78.9 | 2317 | 85.8 | 80.5 | 80.7 | 61.8 | 46.0 | 95.4% |
| PRUNESID | 59.8 | 80.9 | 2218 | 85.9 | 87.6 | 80.4 | 62.4 | 46.3 | **97.0%** |
| Token retention ratio 22.2% (↓ **77.8%**) | | | | | | | | | |
| VisionZip $_{CVPR25}$ | 54.6 | 76.8 | 2224 | 83.4 | 80.4 | 78.5 | 61.3 | 45.2 | 93.2% |
| PRUNESID | 59.0 | 78.0 | 2169 | 85.6 | 86.9 | 78.7 | 61.8 | 45.5 | **95.4%** |
| Token retention ratio 11.1% (↓ **88.9%**) | | | | | | | | | |
| VisionZip $_{CVPR25}$ | 53.2 | 75.8 | 2025 | 78.9 | 80.1 | 73.8 | 58.6 | 44.5 | 89.6% |
| PRUNESID | 55.8 | 73.9 | 2076 | 80.2 | 86.5 | 74.6 | 58.9 | 44.6 | **91.4%** |

94.7%. These results indicate that our compression method maintains strong performance even under large-scene, high-resolution conditions, demonstrating its strong generalization ability beyond standard-resolution benchmarks.

## A.3 ADDITIONAL ABLATION STUDIES

### A.3.1 ABLATION STUDIES ON COMPONENT CONTRIBUTIONS

This section presents detailed ablation studies to isolate and evaluate the individual contributions of two key components in our framework: the Principal Semantic Components Analysis (PSCA) module and the Non-Maximum Suppression (NMS) based redundancy removal mechanism. To accurately assess each component's impact, we design two ablation variants of our model on LLaVA-1.5: 1) *w/o PSCA*: Replaces the PSCA-based grouping with random token grouping, while maintaining the original NMS algorithm. Within each randomly formed group, tokens are sorted by their vector $\ell_2$ norm to determine importance. And 2) *w/o NMS*: Retains the original PSCA grouping and importance ranking, but replaces the NMS-based redundancy removal with a simple top-$k$ selection that ignores token redundancy.

Table 13: Performance comparison of ablation variants on LLaVA-1.5 7B (relative to vanilla baseline, 100%).

| Condition | GQA | MME | POPE | SQA | MMMU | SEED | MMB | Vizwiz | Avg. |
|---|---|---|---|---|---|---|---|---|---|
| *Upper Bound, 576 Tokens (100%)* | | | | | | | | | |
| Vanilla CVPR24 | 61.9 | 1862 | 85.9 | 69.5 | 36.3 | 60.5 | 64.7 | 54.3 | 100% |
| *Retain 192 Tokens ( ↓ 66.7%)* | | | | | | | | | |
| w/o PSCA | 58.8 | 1743 | 84.9 | 68.2 | 35.7 | 58.8 | 62.7 | 54.1 | 97.2% |
| w/o NMS | 59.1 | 1755 | 85.8 | 68.1 | 35.2 | 58.6 | 62.3 | 54.9 | 97.3% |
| Ours | 60.2 | 1797 | 87.1 | 69.1 | 36.8 | 59.0 | 63.8 | 55.5 | **99.3%** |
| *Retain 128 Tokens ( ↓ 77.8%)* | | | | | | | | | |
| w/o PSCA | 57.8 | 1703 | 83.0 | 67.5 | 35.7 | 57.6 | 61.2 | 54.5 | 95.8% |
| w/o NMS | 57.5 | 1722 | 84.9 | 67.4 | 36.1 | 57.4 | 61.4 | 54.3 | 96.3% |
| Ours | 58.9 | 1760 | 86.9 | 68.8 | 35.8 | 57.9 | 62.6 | 56.0 | **98.0%** |
| *Retain 64 Tokens ( ↓ 88.9%)* | | | | | | | | | |
| w/o PSCA | 56.1 | 1670 | 78.5 | 67.3 | 36.1 | 54.5 | 59.3 | 55.2 | 93.9% |
| w/o NMS | 55.9 | 1665 | 79.7 | 67.5 | 36.2 | 54.9 | 58.1 | 56.3 | 94.2% |
| Ours | 57.2 | 1734 | 84.1 | 68.1 | 37.2 | 56.2 | 59.7 | 57.0 | **96.8%** |

Tab. 13 summarizes the comparative performance across all configurations. The experimental results demonstrate that both components contribute significantly to the model's overall performance. Removing either PSCA or NMS leads to consistent performance drops across most datasets and token retention settings. Our full model consistently outperforms both ablation variants across all token retention settings, with an average performance advantage of 1.7-2.9% over the ablation variants, validating the complementary nature of PSCA and NMS.

### A.3.2 ABLATION STUDIES ON INFORMATION PRESERVATION OF BOTH IMPORTANCE AND DIVERSITY

This section presents ablation studies to validate that our method effectively preserves both information importance and diversity of visual tokens. To be specific, we design two ablation experiments on LLaVA-1.5 to isolate these two properties and demonstrate their individual contributions: 1) *Descend*: To test the importance preservation property, we retain tokens in descending order of their importance rankings (i.e., keeping the least important tokens first) while applying our standard NMS for redundancy reduction. This variant directly challenges the importance preservation mechanism by prioritizing less critical information. And 2) *Ascend*: To test the diversity preservation property, we retain tokens in ascending order of their importance rankings (i.e., selecting the most important tokens first) but without NMS to considering redundancy. This variant prioritizes importance while ignoring potential information redundancy and diversity.

Tab. 14 summarizes the performance of all configurations. The *Descend* variant, which retains less informative tokens, suffers severe degradation (e.g., 84.2% at 64 tokens), highlighting the necessity of preserving importance. The *Ascend* variant, which emphasizes importance but ignores redundancy, performs better but notably drops on POPE, underscoring the need for diversity. Our full model consistently surpasses both variants (0.9–1.5% over *Ascend*, 2.7–12.1% over *Descend*), demonstrating its effectiveness in jointly preserving importance and diversity without trade-offs.

### A.3.3 ABLATION STUDIES ON SENSITIVITY TO NMS THRESHOLD SCALING FACTOR $\lambda$

This section presents a comprehensive ablation study to analyze the sensitivity of our method to the NMS threshold scaling factor $\lambda$, and its impact on both model performance and the distribution of retained tokens across semantic groups. As background, our NMS threshold is defined as $\tau = \lambda \cdot \rho$ (where $\rho$ denotes the redundancy metric between tokens), and $\lambda$ is a scaling factor adaptively

Table 14: Performance comparison of importance and diversity preservation ablation variants on LLaVA-1.5 7B (relative to vanilla baseline, 100%).

| Condition | GQA | MME | POPE | SQA | MMMU | SEED | MMB | Vizwiz | Avg (%) |
|---|---|---|---|---|---|---|---|---|---|
| *Upper Bound, 576 Tokens (**100%**)* | | | | | | | | | |
| Vanilla [CVPR24] | 61.9 | 1862 | 85.9 | 69.5 | 36.3 | 60.5 | 64.7 | 54.3 | 100% |
| *Retain 192 Tokens ( ↓ **66.7%**)* | | | | | | | | | |
| *Descend* | 57.3 | 1667 | 83.9 | 66.8 | 35.8 | 56.4 | 58.8 | 54.8 | 96.1% |
| *Ascend* | 59.1 | 1763 | 85.8 | 68.1 | 36.2 | 58.6 | 63.3 | 54.9 | 97.9% |
| Ours | 60.1 | 1791 | 86.9 | 68.5 | 36.1 | 59.0 | 63.7 | 55.4 | **98.8%** |
| *Retain 128 Tokens ( ↓ **77.8%**)* | | | | | | | | | |
| *Descend* | 56.0 | 1556 | 79.6 | 66.4 | 35.8 | 53.8 | 55.5 | 54.9 | 92.1% |
| *Ascend* | 57.5 | 1722 | 84.9 | 68.4 | 36.1 | 57.4 | 62.4 | 55.3 | 96.9% |
| Ours | 58.8 | 1749 | 86.5 | 68.3 | 35.8 | 57.8 | 62.1 | 55.8 | **97.6%** |
| *Retain 64 Tokens ( ↓ **88.9%**)* | | | | | | | | | |
| *Descend* | 51.7 | 1361 | 67.5 | 64.8 | 35.1 | 49.3 | 43.6 | 54.1 | 84.2% |
| *Ascend* | 55.9 | 1664 | 79.7 | 68.5 | 36.2 | 54.9 | 58.1 | 56.3 | 94.8% |
| Ours | 57.1 | 1733 | 83.8 | 67.8 | 37.0 | 56.1 | 58.8 | 56.9 | **96.3%** |

Table 15: Performance sensitivity to NMS threshold scaling factor $\lambda$ on LLaVA-1.5 7B (relative to vanilla baseline, 100%).

| $\alpha$ | Retain 64 | | | | | Retain 128 | | | | | Retain 192 | | | | |
|---|---|---|---|---|---|---|---|---|---|---|---|---|---|---|---|
| | GQA | MME | POPE | SQA | Avg. | GQA | MME | POPE | SQA | Avg. | GQA | MME | POPE | SQA | Avg. |
| 24 | 56.7 | 1726 | 83.4 | 68.0 | 94.8% | 58.4 | 1718 | 85.5 | 68.3 | 96.1% | 59.7 | 1770 | 86.3 | 68.4 | 97.6% |
| 28 | 57.0 | 1718 | 84.1 | 67.8 | 94.9% | 58.8 | 1723 | 86.4 | 68.2 | 96.6% | 59.9 | 1783 | 86.8 | 68.4 | 98.0% |
| 32 | 57.1 | 1733 | 83.8 | 67.8 | **95.1%** | 58.8 | 1749 | 86.5 | 68.3 | **97.0%** | 60.1 | 1791 | 86.9 | 68.5 | **98.3%** |
| 36 | 57.3 | 1705 | 84.7 | 67.9 | 95.1% | 58.7 | 1733 | 85.9 | 68.3 | 96.5% | 59.7 | 1789 | 86.6 | 68.5 | 98.0% |
| 40 | 57.3 | 1675 | 84.7 | 67.9 | 94.7% | 58.6 | 1711 | 85.7 | 68.1 | 96.2% | 59.6 | 1772 | 86.3 | 68.4 | 97.6% |

determined by the global token budget $N$ (with a default setting of $\lambda = \frac{N}{32}$ in our base model). We design two complementary experiments to evaluate the role of $\alpha$ with $\lambda = \frac{N}{\alpha}$:

**Performance Sensitivity.** We test distinct values of $\alpha$ (24, 28, 32, 36, 40) under three token retention settings (192, 128 and 64 tokens) on the LLaVA-1.5 7B model. We report performance across four key benchmarks (GQA, MME, POPE, SQA) to quantify how $\lambda$ impacts multimodal understanding. Tab. 15 summarizes the performance across different $\alpha$ values. The results demonstrate that performance remains highly stable across different values of $\alpha$ (24–40), indicating that the method is not sensitive to fine-tuning of this parameter. The optimal performance is consistently achieved at $\alpha = 32$, yielding the highest average scores across multiple token retention settings.

**Token Distribution Sensitivity.** For the 64-token retention setting (on the MME benchmark), analyze how $\alpha$ affects the number of retained tokens across 16 semantic groups (Group 0 to Group 15) generated by the PSCA module. We also include a "Before pruning" baseline to show the initial token count per group prior to NMS-based redundancy removal. As shown in Tab. 16, we can observe that the distribution of group token counts remains relatively stable across different values of $\alpha$. Additionally, we note the following observations and explanations:

- *The number of tokens retained in Group 0 is small:* Group ids are related to the principal component ordering in PSCA. Group 0 often corresponds to the background regions of the image that exhibit higher redundancy and contain less important semantic information, resulting in a smaller number of retained tokens.

- *The number of tokens retained increases as group id increases:* According to the nature of PCA, later principal component groups are associated with tokens that have greater variability and lower redundancy. Under the same NMS threshold, more tokens are retained after redundancy removal.

Table 16: Token distribution across PSCA groups (Group 0–15) for different $\lambda$ (retain 64 tokens, MME benchmark).

| Group id $\alpha$ | 0 | 1 | 2 | 3 | 4 | 5 | 6 | 7 | 8 | 9 | 10 | 11 | 12 | 13 | 14 | 15 |
|---|---|---|---|---|---|---|---|---|---|---|---|---|---|---|---|---|
| Before pruning | 153.6 | 28.1 | 37.5 | 36.3 | 34.4 | 31.1 | 28.1 | 26.8 | 25.1 | 23.9 | 23.3 | 23.1 | 23.9 | 25.5 | 27.1 | 28.4 |
| 24 | 1.0 | 4.2 | 4.1 | 3.8 | 3.8 | 4.0 | 4.1 | 4.1 | 4.1 | 4.0 | 4.0 | 4.0 | 4.2 | 4.4 | 4.6 | 4.8 |
| 28 | 1.1 | 4.3 | 4.3 | 4.1 | 4.0 | 4.2 | 4.1 | 4.1 | 4.0 | 3.8 | 3.8 | 3.8 | 4.0 | 4.2 | 4.4 | 4.6 |
| 32 | 1.1 | 3.8 | 3.8 | 3.6 | 3.7 | 4.0 | 4.1 | 4.1 | 4.2 | 4.1 | 4.1 | 4.1 | 4.3 | 4.5 | 4.7 | 4.9 |
| 36 | 1.1 | 3.4 | 3.5 | 3.5 | 3.6 | 3.9 | 4.1 | 4.1 | 4.2 | 4.2 | 4.2 | 4.3 | 4.4 | 4.6 | 4.8 | 5.0 |
| 40 | 1.3 | 3.0 | 3.3 | 3.4 | 3.6 | 3.8 | 4.0 | 4.1 | 4.2 | 4.2 | 4.3 | 4.4 | 4.6 | 4.8 | 5.0 | 5.2 |

- *As $\alpha$ increases, tokens are more likely to be distributed in higher-numbered groups:* As $\alpha$ increases, $\lambda$ decreases, and the NMS redundancy removal threshold $\tau = \lambda \cdot \rho$ also lowers. Tokens in lower-ranked, more redundant groups are more likely to be pruned, leading to a smaller number of retained tokens.

These findings provide further insight into how $\alpha$ influences the distribution of tokens across different groups, contributing to the overall balance between redundancy removal and token retention.

## A.4 VISUALIZATION

### A.4.1 VISUALIZATION OF PSCA GROUPING AND RETAINED TOKENS

We present additional visualizations of PSCA-based token grouping and the final retained tokens to demonstrate the effectiveness of our method. Fig. 8 shows partial results of our semantic token grouping and compares the retained tokens between our method and the attention-guided baseline, VisionZip, under an extreme compression ratio (retaining 32 tokens, 5.6%). Fig. 9 shows the grouping and selection results when retaining 64 tokens (11.1%). As observed, our method selectively preserves representative tokens from highly redundant groups with shared semantics, enabling broader semantic diversity within a limited token budget. In contrast, VisionZip tends to retain tokens with high attention scores that are often semantically similar, leading to redundancy and insufficient coverage of diverse visual concepts.

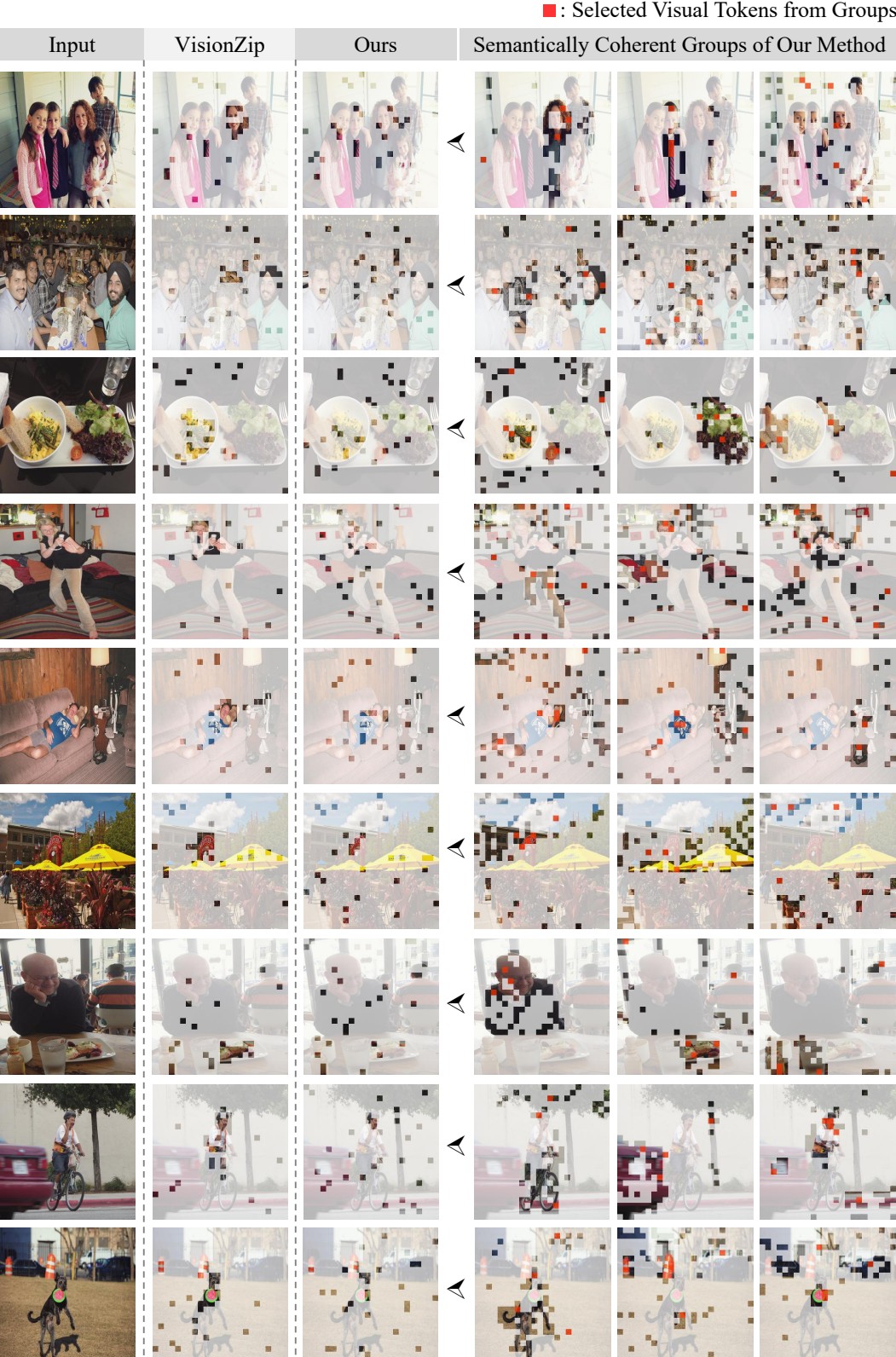

Figure 8: Visualization of PSCA-based semantic token grouping and final retained tokens under an extreme compression ratio (retaining 32 tokens, 5.6%). *Semantically Coherent Groups* show partial grouping results from our PSCA stage, and *Retained* compares the selected tokens from our method and VisionZip. Our approach preserves representative tokens from semantically redundant groups, enabling broader information coverage than attention-only methods like VisionZip.

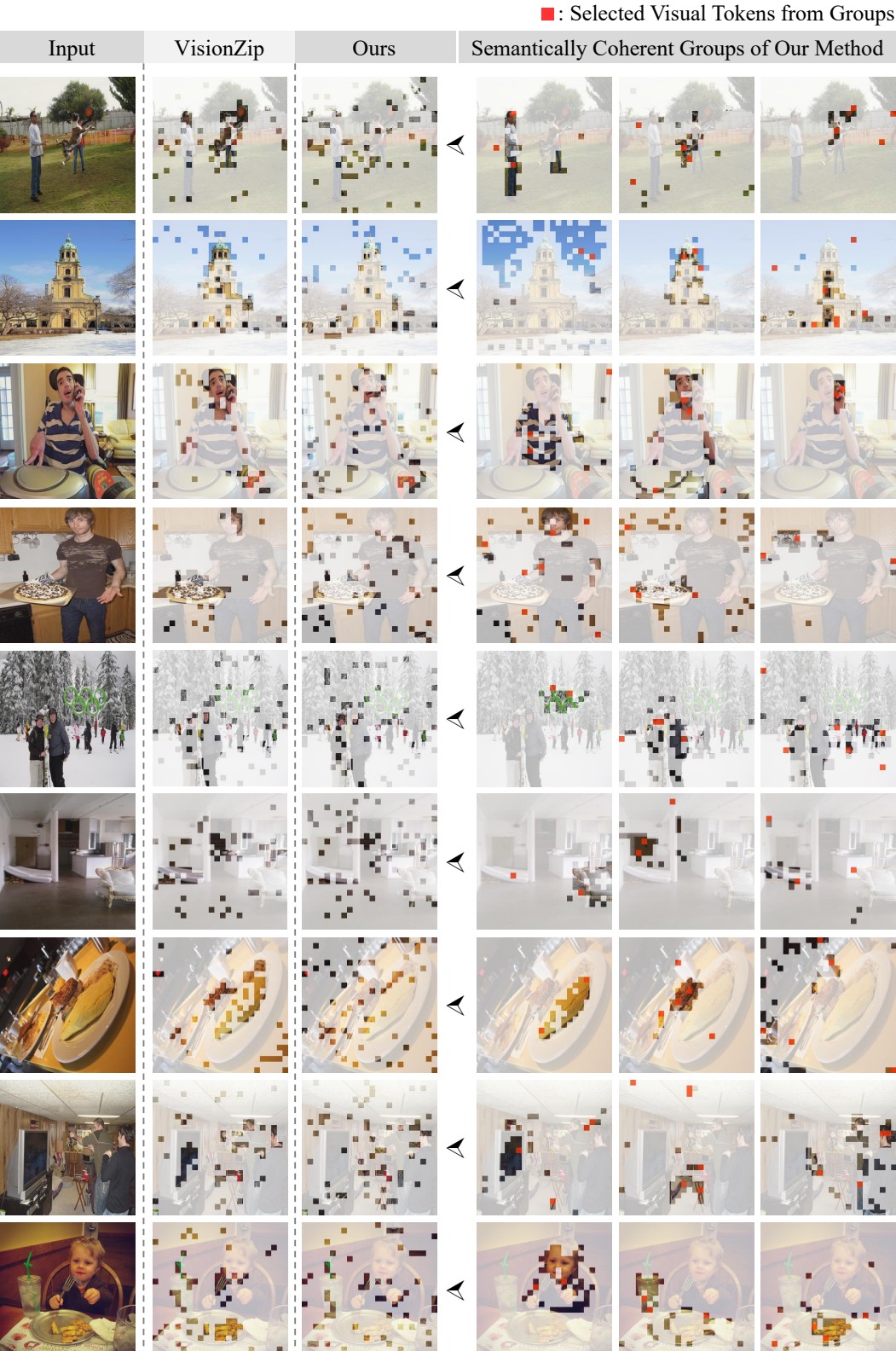

Figure 9: Visualization of PSCA-based token grouping and final retained tokens when retaining 64 tokens (11.1%). Similar to Fig. 8, our method effectively selects representative tokens across diverse groups while filtering redundancy. This helps maintain semantic diversity under tight token budgets, outperforming redundancy-prone baselines like VisionZip.

