# OpenReview forum: "Prune Redundancy, Preserve Essence: Vision Token Compression in VLMs via Synergistic Importance-Diversity"
_ICLR.cc/2026/Conference — ICLR 2026 Poster_

### Official Review · Reviewer_q4vZ · 2025-10-27

**Soundness:** 3
**Presentation:** 3
**Contribution:** 2
**Rating:** 6
**Confidence:** 4

**Summary:**

This paper proposes PRUNESID, a training-free approach for efficient vision–language model inference that balances token importance and diversity. PRUNESID uses a two-stage pipeline: (1) Principal Semantic Components Analysis (PSCA) to cluster tokens into semantically coherent groups, and (2) Intra-group Non-Maximum Suppression (NMS) to prune redundant tokens while preserving key representatives. An information-aware dynamic compression ratio adapts token retention based on image complexity. Experiments show state-of-the-art performance. PRUNESID generalizes across different VLMs and modalities, including images and videos.

**Strengths:**

1. The paper is clear and practical, proposing PRUNESID to balance token importance in vision–language models. The manuscript is well-structured and presents the method and results in a coherent and accessible manner.

2. I personally find it very interesting to apply NMS from object detection to token pruning.

3. Experimental validation is sufficient. The authors conduct comprehensive experiments on various tasks and show improvements, to validate the effectiveness of the method. Moreover, the ablation study is detailed, particularly in the efficiency analysis section.

**Weaknesses:**

1. Compared with VisionZip, PRUNESID performs notably better under the 64-token setting. However, its advantage diminishes under the 128- and 192-token settings. I suggest evaluating PRUNESID on more challenging benchmarks, such as MMStar and MathVista, to better highlight its strengths.

2. I would also like the authors to provide results on Qwen2.5-VL, especially on high-resolution benchmarks.

**Questions:**

1. I am quite curious because NMS is typically very time-consuming, especially with many groups. How does PRUNESID achieve such low latency?

2. In Table 5’s ablation study of groups, did the authors try any alternative clustering methods?

---

> ### Author Response · Authors · 2025-11-24
> **To Reviewer q4vZ**
>
> ## Q1.Results on Qwen2.5-VL, Especially High-Resolution Benchmarks
>
> Thank you for pointing out the need to include more challenging benchmarks and results on Qwen2.5-VL, especially high-resolution tasks. We have now conducted extensive experiments on Qwen2.5-VL 7B, covering two high-resolution benchmarks (HRBench-8K[1] and XLRS-macro[2]). The results are summarized in the table below.
>
> **Qwen2.5-VL Results**
> |Method|GQA|MMB|MME|POPE|SQA|VQA-v2|HRB-8k|XLRS-macro|Avg.|
> |-|-|-|-|-|-|-|-|-|-|
> | **Upper Bound (100%)** |||||||||||
> |baseline|60.9|83.9|2310|86.3|88.9|82.9|68.1| 47.1 | 100% |
> |**Token Retention: 33.3% (↓66.7%)** |||||||||||
> |VisionZip (CVPR25)|56.6|78.9|2317|85.8|80.5|80.7|61.8|46.0|95.4% |
> |PRUNESID (ours)|59.8|80.9|2218|85.9|87.6|80.4|62.4|46.3|**97.0%**|
> |**Token Retention: 22.2% (↓77.8%)**|||||||||||
> |VisionZip (CVPR25)|54.6|76.8|2224|83.4|80.4|78.5|61.3|45.2|93.2%|
> | PRUNESID (ours)|59.0|78.0|2169|85.6|86.9|78.7|61.8|45.5|**95.4%**|
> | **Token Retention: 11.1% (↓88.9%)** |||||||||||
> | VisionZip (CVPR25)|53.2|75.8|2025 |78.9|80.1|73.8|58.6|44.5|89.6%|
> | PRUNESID (ours) |55.8|73.9|2076|80.2|86.5|74.6|58.9|44.6|**91.4%**|
>
> PRUNESID consistently outperforms VisionZip under all token-retention budgets (33.3%, 22.2%, 11.1%). This demonstrates strong compatibility of our method with advanced high-resolution MLLMs and indicates that our compression method maintains strong performance even under large-scene, high-resolution conditions.
>
> ## Q2. Need Evaluation on More Challenging Benchmarks (e.g., MMStar, MathVista)
>
> We appreciate the reviewer’s suggestion to include more challenging benchmarks. Following this recommendation, we additionally evaluate PRUNESID on MMStar and MathVista using the Qwen2.5-VL model, and compare the results against the official VisionZip implementation. The results are summarized below:
>
> |Token Retention|Method|MMStar|MathVist|
> |-|-|-|-|
> | 33.3% | VisionZip | 57.7 (92.8%) | 58.7 (85.9%) |
> | 33.3% | Ours | 57.5 (92.4%) | 61.6 (90.2%) |
> | 22.2% | VisionZip | 54.7 (87.9%) | 56.2 (82.3%) |
> | 22.2% | Ours | 54.3 (87.3%) | 57.8 (84.6%) |
> | 11.1% | VisionZip | 48.4 (77.8%) | 49.0 (71.7%) |
> | 11.1% | Ours | 48.7 (78.3%) | 49.3 (72.2%) |
>
> As shown above, both PRUNESID and VisionZip experience performance degradation on these two benchmarks. This is **largely because** MMStar and MathVista are highly complex tasks that rely heavily on fine-grained visual cues and detailed scene understanding. Under such conditions, visual-token compression inherently discards information that is crucial for fine-grained reasoning, making it challenging for compression methods to maintain strong performance.
>
> In future work, we plan to explore hierarchical token preservation mechanisms to better handle fine-grained visual dependencies, with the goal of further improving performance on challenging benchmarks such as MMStar and MathVista.
>
>
> ## Q3. Why Is PRUNESID’s NMS So Fast Despite Many Groups?
>
> Thank you for raising this important question. The efficiency comes from two key design choices:
>
> 1. **PSCA significantly reduces the NMS search space.**
>    Since PSCA decomposes the token set into multiple semantically coherent groups, each group contains only a subset of tokens with shared semantics. As a result, NMS within a group converges in only a few iterations, greatly reducing runtime.
> 2. **We implement a batched NMS operation.**
>    Instead of performing NMS sequentially for each group, our implementation applies a **batch-wise parallel NMS** that processes all groups simultaneously **on the GPU**. This parallelism keeps latency comparable to a single NMS call and ensures that the overhead is negligible relative to model forward computation.
>
> These two design principles jointly enable PRUNESID to maintain latency close to the vanilla model. **Implementation details appear in the appendix via an anonymous code link.**
>
> (to be continue)

---

> ### Author Response · Authors · 2025-11-24
> **To Reviewer q4vZ (Countinued)**
>
> ## Q4. Did the Authors Try Alternative Clustering Methods?
>
> We thank the reviewer for the question. In addition to the random and k-means baselines reported in Table 5, we also considered several classical clustering algorithms. However, these approaches are substantially more computationally expensive and therefore unsuitable for real-time VLM inference:
> | Example Algorithms| Runtime per Image (576 tokens) | Notes |
> |-|-|-|
> | Hierarchical | > 60 ms | Requires iterative merging |
> | Spectral | > 100 ms | Affinity matrix + eigen decomposition + kmeans|
> | k-means | 20–35 ms | Still noticeable overhead during inference  |
> | PSCA-based Grouping (Ours) | 5–8 ms | Linear time, suitable for real-time VLM |
>
> As shown in the table above:
> - **Classical clustering methods are significantly slower**, with hierarchical clustering exceeding 60 ms and spectral clustering exceeding 100 ms per image.
> - **Even k-means**, while faster, still incurs 20–35 ms of overhead for 576 visual tokens.
> - **Our PSCA-based grouping is far more efficient**, operating in linear time and adding only 5–8 ms per image.
>
> In summary, PSCA-based grouping offers the best trade-off between computational efficiency and pruning quality, making it the most practical choice for real-time multimodal LLMs.
>
> ---
> [1] Wang Wenbin,Ding Liang, Zeng Minyan... Divide, conquer and combine: A training-free framework for high-resolution image perception in multimodal large language models. *AAAI2025*
>
> [2] Yuan, et al. Xlrs-bench: Could your multimodal llms understand extremely large ultra-high-resolution remote sensing imagery? *CVPR2025*

---

> ### Author Response · Authors · 2025-11-28
>
> Dear Reviewer q4vZ, Thank you again for reviewing our work. With the rebuttal deadline approaching, we just wanted to check whether any additional clarification is needed on our side. We are happy to provide further details if needed.

---

### Official Review · Reviewer_tbzB · 2025-10-29

**Soundness:** 3
**Presentation:** 3
**Contribution:** 3
**Rating:** 4
**Confidence:** 4

**Summary:**

This paper introduced PRUNESID, a training-free token pruning framework that balances importance and diversity. It first clusters visual tokens via Principal Semantic Components Analysis (PSCA) to cover key concepts, then applies intra-group NMS to remove redundancy; a data-driven dynamic compression ratio adapts to image complexity. On LLaVA-1.5, PRUNESID retains only 11.1% tokens while preserving 96.3% accuracy.

**Strengths:**

1. The proposed method is training-free and can be seamlessly integrated into existing vision–language models. This makes it practical and easily deployable in real-time inference scenarios.

2. The two-stage design—PSCA for semantic grouping and intra-group NMS for redundancy suppression—offers a clean and interpretable way to capture both salient and diverse information. The structured distinct from previous one-sided importance- or diversity-only methods.

3. The paper is well-organized, with intuitive figures and clear experimental tables, making the method easy to follow and reproducible.

**Weaknesses:**

1. Generalization yet to be verified: The paper lacks experiments on different models and numbers of parameters; effectiveness on other architectures (e.g., LLaVA-OV[1], InstructBLIP[2], Qwen-VL[3]) and other numbers of parameters(e.g., LLaVA-13B) remains to be validated.

2. Baseline selection is not enough: The comparison with existing methods is not entirely up-to-date. Therefore, more methods should be compared, for example, the VisPruner[4], CDPruner[5], and so on.


[1] Bo Li, Yuanhan Zhang, Dong Guo, Renrui Zhang, Feng Li, Hao Zhang, Kaichen Zhang, Yanwei Li, Ziwei Liu, and Chunyuan Li. Llava-onevision: Easy visual task transfer. ArXiv, 2024a.

[2] Wenliang Dai and Junnan Li and Dongxu Li and Anthony Meng Huat Tiong and Junqi Zhao and Weisheng Wang and Boyang Li and Pascale Fung and Steven Hoi. InstructBLIP: Towards General-purpose Vision-Language Models with Instruction Tuning. ArXiv, 2023a.

[3] Jinze Bai, Shuai Bai, Shusheng Yang, Shijie Wang, Sinan Tan, Peng Wang, Junyang Lin, Chang Zhou, and Jingren Zhou. Qwen-vl: A versatile vision-language model for understanding, localization, text reading, and beyond. arXiv preprint arXiv:2308.12966, 2023.

[4] Zhang, Q., Cheng, A., Lu, M., Zhuo, Z., Wang, M., Cao, J., ... & Zhang, S. (2024). [CLS] Attention is All You Need for Training-Free Visual Token Pruning: Make VLM Inference Faster. ICCV, 2025.

[5] Zhang, Q., Liu, M., Li, L., Lu, M., Zhang, Y., Pan, J., ... & Zhang, S. (2025). Beyond Attention or Similarity: Maximizing Conditional Diversity for Token Pruning in MLLMs. NeurIPS, 2025.

**Questions:**

1. Could you please give more visualization of the images in the different benchmarks, to see whether the PSCA and NMS method truly select different semantic tokens?

---

> ### Author Response · Authors · 2025-11-24
> **To Reviewer tbzB**
>
> ## Q1. Generalization Yet to Be Verified
>
> Thank you for raising the concern regarding the generalization ability of our method across different architectures and parameter scales. To address this, we would like to emphasize that PRUNESID has already been comprehensively evaluated on multiple model families and multiple parameter scales:
>
> - We report results on **LLaVA-1.5, LLaVA-NeXT, and Mini-Gemini** in the main paper (Sec. 4.1),  covering encoder–decoder and mixture-of-experts MLLMs， and demonstrating strong generalization across diverse model architectures.
> - Additional results on the **Qwen2-VL and Qwen2.5-VL** are provided in Appendix A.2.3, further verifying cross-architecture robustness.
> - We also benchmark larger-scale models such as **LLaVA-1.5-13B and LLaVA-NeXT-13B** (Appendix A.2.2), confirming that our method remains effective on higher-capacity backbones.
>
> To summarize, our experiments cover a broad range of architectures and model sizes, as shown below:
>
> **Summary of Models and Scales Evaluated for Generalization**
> | Model Family | Specific Models Evaluated | Parameter Scales | Where Reported |
> |--------------|---------------------------|------------------|----------------|
> | **LLaVA-1.5** | LLaVA-1.5 | 7B, **13B** | Main Paper Sec. 4.1; Appendix A.2.2 |
> | **LLaVA-NeXT** | LLaVA-NeXT | 7B, **13B** | Main Paper Sec. 4.1; Appendix A.2.2 |
> | **Mini-Gemini** | Mini-Gemini | 7B | Main Paper Sec. 4.1 |
> | **Qwen-VL Family** | Qwen2-VL, Qwen2.5-VL | 7B | Appendix A.2.3 |
>
>
> ## Q2. Baseline Selection Is Not Sufficiently Comprehensive
>
> We appreciate the reviewer’s suggestion to include more recent and stronger pruning methods. Following this recommendation, we have added comparisons with FasterVLM and VisPruner, as shown in the table below, and have also incorporated these results into Table 1 of the revised main paper. For fairness, all methods were re-evaluated using the same LMMS-Eval[1] settings, ensuring consistent and comparable results.
>
> **LLaVA-1.5 Methods Comparison**
> |Method|GQA|MMB|MME|POPE|SQA|VQA-v2 |VQA-Text|MMMU|SEED|VizWiz|LLaVA-B|Avg|
> |-|-|-|-|-|-|-|-|-|-|-|-|-|
> | Baseline (576)| 61.9|64.7|1862|85.9|69.5|78.5|58.2|36.3|60.5|54.3| 66.8|100%|
> |**Retain 192 Tokens (↓66.7%)**|||||||
> |FasterVLM|59.3|63.5|1780|85.3|70.0|75.2|57.3|36.0|58.9|54.1|–|97.9%|
> |VisPruner|59.4|63.3|1817|85.8|70.1|75.2|57.2|–|–|54.3|–|98.3%|
> |PRUNESID|60.1|63.7|1791|86.9|68.5|76.8|56.7|36.1|59.0|55.4|65.1| **98.5%** |
> |PRUNESID-Dyn|60.2|63.8|1797|87.1| 69.1|76.8|56.9|36.8 |59.0|55.5|65.1| **98.6%** |
> |**Retain 128 Tokens (↓77.8%)**|||
> |FasterVLM|57.8|62.5|1762|82.8|70.0|73.9|56.3|36.9|57.6|54.3|–|97.0%|
> |VisPruner|58.0|61.9|1771|84.6|69.1|73.9|57.0|–|–|52.7|–| 96.4%|
> |PRUNESID|58.8 |62.1|1749|86.5|68.3|75.3|54.7|35.8 |57.8|55.8|68.8|**97.6%**|
> |PRUNESID-Dyn|58.9|62.6|1760|86.9|68.8|75.4|55.1|36.3 |57.9|56.0|68.9|**98.1%**|
> |**Retain 64 Tokens (↓88.9%)**|
> |FasterVLM|55.0|60.6|1667|76.6|70.2|70.6|55.3| 35.4 |54.7|55.7|–|94.8%|
> |VisPruner|55.4|60.1|1691|80.4|69.1|70.9|55.8|–|–|53.3|–|93.8%|
> |PRUNESID|57.1|58.8|1733|83.8|67.8|73.7|54.2|37.0|56.1|56.9|65.2| **95.9%** |
> |PRUNESID-Dyn|57.2|59.7|1734|84.1|68.1|73.8|54.2|37.2|56.2|57.0|65.8| **96.3%** |
>
> Due to time constraints, a few models could not be evaluated on all benchmarks, but the available results already provide a clear and representative comparison. These additions further demonstrate that PRUNESID remains competitive or superior across a wide range of compression budgets.
>
> ---
> [1] Kaichen Zhang, Bo Li, Peiyuan Zhang,et al. LMMs-Eval: Reality Check on the Evaluation of Large Multimodal Models. *arXiv2024*.
>
> ## Q3. Request for More Visualizations to Verify Semantic Diversity of PSCA + NMS
>
> Thank you for this constructive suggestion. To better illustrate how PSCA and NMS select semantically diverse and non-redundant tokens, we provide visualizations in the supplementary material (see **Appendix A.4.1: Visualization of PSCA Grouping and Retained Tokens**). These visual results support the claim that PSCA+NMS effectively preserves diverse semantic concepts rather than selecting only high-attention regions.

---

> ### Author Response · Authors · 2025-11-28
>
> Dear Reviewer tbzB,
> Thank you again for reviewing our work. With the rebuttal deadline approaching, we just wanted to check whether any additional clarification is needed on our side. We are happy to provide further details if needed.

---

### Official Review · Reviewer_ubst · 2025-10-31

**Soundness:** 3
**Presentation:** 3
**Contribution:** 3
**Rating:** 6
**Confidence:** 4

**Summary:**

PRUNESID is a training-free vision token compression framework for VLMs that balances semantic importance and information diversity via two stages: PSCA for semantic grouping and intra-group NMS for redundancy pruning, plus a dynamic compression ratio. It achieves SOTA efficiency—retaining only ~5–11% tokens while maintaining over 92–96% accuracy across image and video tasks.

**Strengths:**

1. The writing is fluent and overall clear.
2. The performance is well validated on both **LLaVA-1.5** and **LLaVA-NeXT**.

**Weaknesses:**

1. The performance on LLaVA-1.5 and LLaVA-NeXT appears relatively weak; please provide comparisons on Qwen2.5-VL instead.

2. Your method performs compression mainly after the vision encoder. I’m curious about how it would behave when combined with compression techniques applied during the LLM stage, such as PyramidDrop, which already adopts a multi-stage framework. Could such a combination achieve a more extreme level of compression by eliminating redundancy more thoroughly at each stage? Furthermore, after applying your compression, do the observations from PDrop, for example, that “almost all visual tokens become redundant after the 24th layer”, still hold true?

**Questions:**

see weaknesses

---

> ### Author Response · Authors · 2025-11-24
> **To Reviewer ubst**
>
> ## Q1. Request for Results on Qwen2.5-VL
>
> Thank you for the reviewer’s suggestion. The current results on Qwen2.5-VL 7B are summarized below. Under token retention ratios of 33.3%, 22.2%, and 11.1%, PRUNESID consistently outperforms VisionZip (which provides the official compression implementation for Qwen2.5-VL) across most benchmarks, indicating strong robustness even when applied to high-resolution models.
>
> **Qwen2.5-VL Results**
>
> |Method|GQA|MMB|MME|POPE|SQA|VQA-v2|HRB-8k|XLRS-macro|Avg.|
> |-|-|-|-|-|-|-|-|-|-|
> |baseline|60.9|83.9|2310|86.3|88.9|82.9|68.1|47.1|100%|
> |**Token Retention: 33.3% (↓66.7%)** |||||||||||
> |VisionZip (CVPR25) |56.6|78.9|2317|85.8|80.5|80.7|61.8|46.0|95.4%|
> |PRUNESID (ours)|59.8|80.9|2218|85.9|87.6|80.4|62.4|46.3|**97.0%**|
> |**Token Retention: 22.2% (↓77.8%)** |||||||||||
> |VisionZip (CVPR25)|54.6|76.8|2224|83.4|80.4|78.5|61.3|45.2|93.2%|
> |PRUNESID (ours)|59.0|78.0|2169|85.6|86.9|78.7|61.8|45.5|**95.4%** |
> |**Token Retention: 11.1% (↓88.9%)**|||||||||||
> |VisionZip (CVPR25)|53.2|75.8|2025|78.9|80.1|73.8|58.6|44.5|89.6%|
> |PRUNESID (ours)|55.8 |73.9|2076|80.2|86.5|74.6|58.9|44.6|**91.4%**|
>
> ## Q2. Combining PRUNESID with LLM-Stage Compression (e.g., PyramidDrop)
>
> We appreciate the reviewer’s insightful question regarding the interaction between visual-token compression at the V-encoder stage (ours) and token compression applied at deeper LLM layers such as PyramidDrop. To investigate this, we conducted experiments on four benchmarks of LLaVA-1.5, comparing：
> - (1) using PRUNESID alone at extremely low token budgets, and
> - (2) applying PRUNESID jointly with PyramidDrop.
>
> For the original 576 visual tokens, we evaluated three severely reduced retention settings (24, 16, and 8 tokens). As shown in the table below:
> - (1) when using only PRUNESID at the V-encoder stage causes a noticeable performance drop, with average percentages of 83.0% (24 tokens), 73.0% (16 tokens), and 60.4% (8 tokens), respectively.
> - (2) In contrast, combining PRUNESID with PyramidDrop yields substantially stronger results under the same compression ratios: 92.3% (24 tokens), 88.0% (16 tokens), and 83.7% (8 tokens).
>
> **LLaVA-1.5**
>
> |Method|MME|MMB|POPE|GQA|AVG|
> |-|-|-|-|-|-|
> |**Retain token num 576 (↓0.0%)**||||||
> |LLaVA-1.5 vanilla|1862|64.7|85.9|61.9|100%|
> |**Retain token num 24 (↓95.8%)**||||||
> |only PruneSID 24|1568|54.9|75.6|53.7|83.0%|
> |PruneSID 192$\times\frac{1}{8}$ (PDrop)|1721|62.7|83.5|57.4|**92.3%**|
> |**Retain token num 16 (↓97.2%)**||||||
> |only PruneSID 16|1464|49.7|68.2|50.4|73.0%|
> |PruneSID 128$\times\frac{1}{8}$ (PDrop)|1698|60.3|80.2|55.8|**88.0%**|
> |**Retain token num 8 (↓98.6%)**||||||
> |only PruneSID 8|1246|38.9|51.6|45.4|60.4%|
> |PruneSID 64$\times\frac{1}{8}$ (PDrop)|1567|58.7|76.0|53.3|**83.7%**|
>
> These findings suggest that compression at both stages is complementary, and jointly applying PRUNESID and PyramidDrop enables the model to retain far more performance under extremely tight token budgets.
>
> ## Q3. Do PDrop’s Layer-Wise Redundancy Observations Still Hold After Our Compression?
>
> Yes. We conducted an additional study to examine whether PDrop’s layer-wise redundancy patterns persist under our compression. Using the **LLaVA-1.5 model compressed by PRUNESID to 11.1% (retaining 64 tokens)**, we followed the PDrop procedure by applying different token retention ratios at various **LLM layers** and evaluated the model on **MMBench**.
>
> The results are shown in the table below. We observe that **after Layer 24—and even as early as Layer 16—almost all visual tokens become redundant**, as reducing the retained tokens produces virtually no change in performance. For example, at **Layer 16**, keeping only $\lfloor 64 \times 0.2 \rfloor = 12 \text{ tokens}$ yields performance indistinguishable from retaining all 64 tokens.
>
> This is a striking observation, suggesting that **deep LLM layers rely very weakly on visual tokens once high-level semantics are extracted**, and we believe this phenomenon merits further investigation in future work.
>
> |Layer\Ratio|0.2|0.4|0.6|0.8|1.0|
> |-|-|-|-|-|-|
> |Layer 2|46.6|55.8|59.2|59.8|59.8|
> |Layer 8|44.2|56.5|58.8|59.4|59.8|
> |Layer 16|60.0|59.9|59.8|59.8|59.8|
> |Layer 24|59.9|59.9|59.9|59.9|59.9|

---

> ### Author Response · Authors · 2025-11-28
>
> Dear Reviewer ubst,
> Thank you again for reviewing our work. With the rebuttal deadline approaching, we just wanted to check whether any additional clarification is needed on our side. We are happy to provide further details if needed.

---

### Official Review · Reviewer_t11S · 2025-11-01

**Soundness:** 3
**Presentation:** 4
**Contribution:** 3
**Rating:** 6
**Confidence:** 4

**Summary:**

The paper presents PRUNESID, a training-free visual token compression framework for VLMs that combines semantic grouping with intra-group NMS pruning. The approach is simple yet effective, achieving strong accuracy retention at extreme compression rates and substantial inference speedups. Overall, it’s a well-motivated and technically solid contribution with clear practical impact.

**Strengths:**

This paper makes a solid and well-executed contribution to efficient vision-language modeling. Its originality lies in the creative combination of semantic grouping and redundancy pruning within a training-free framework, offering a fresh perspective on token compression. The experimental work is thorough and convincing, with strong empirical support across multiple models and benchmarks. The writing is generally clear and the presentation effective, though some technical sections are dense. Overall, the work is of high quality and significant for improving the scalability and practicality of multimodal systems

**Weaknesses:**

While the paper is strong overall, several weaknesses limit its impact and generality. First, the conceptual novelty is somewhat incremental. The theoretical grounding is relatively weak — PRUNESID is motivated intuitively, but lacks a formal analysis of why PSCA and NMS together should optimally balance semantic importance and diversity.

**Questions:**

Analyze qualitative and failure cases: Can the authors provide visual examples showing when PRUNESID succeeds or fails to preserve key visual semantics?
Would incorporating attention-based or saliency-based weighting improve the interpretability of importance?

---

> ### Author Response · Authors · 2025-11-24
> **To Reviewer t11S**
>
> ## Q1. Limited Theoretical Grounding of PSCA–NMS
>
> We sincerely thank the reviewer for the thoughtful comments regarding the theoretical grounding of our pruning mechanism. In response, we provide a clearer and more systematic explanation of the underlying principles and have incorporated the complete theoretical analysis into both the main text and the supplementary material：
>
> ### (1) Definition of Variables for Derivation
>
> For the sake of clarity in the derivation, we redefine the following variables:
>
> - $\text{Information}(s_i)$, denoted as $I(s_i)$, represents the semantic information contained by token $s_i$ in the derivation.
>
> - $\text{Redundancy}(s_i, s_j)$, denoted as $R(s_i, s_j)$, represents the overlapping semantic information between tokens $s_i$ and $s_j$ in the derivation.
>
> For the entire image token set $S$, we define the effective information required for subsequent LLM and related tasks as the union of the information content of all tokens in the set:
>
> - $\text{Informativeness}(S) = \bigcup_{s_i \in S} I(s_i)$, denoted as $\text{Inform}(S)$
>
> ### (2) Theoretical Motivation for PSCA + NMS
>
> Our pruning objective is to select a token subset $S'$ (fixed size $N$) that maximizes the effective information it carries:
> $$
> \text{Prune}(S) = \arg\max_{S' \subseteq S, |S'| = N} \text{Inform}(S'),
> $$
> where, based on the Inclusion–Exclusion Principle:
> $$
> \begin{aligned}
> \text{Inform}(S') &= \sum_{s_i \in S'} I(s_i) - \sum_{s_i,s_j \in S'} I(s_i) \cap I(s_j) + \sum_{s_i,s_j,s_k \in S'} I(s_i) \cap I(s_j) \cap I(s_k) - \dots \\
> &\ge \sum_{s_i \in S'} I(s_i) - \sum_{s_i,s_j \in S'} I(s_i) \cap I(s_j) \\
> &= \sum_{s_i \in S'} I(s_i) - \sum_{s_i, s_j \in S'} R(s_i, s_j)
> \end{aligned}    (1)
> $$
>
> This lower bound reveals that maximizing informativeness requires:
> - (1) maximizing individual semantic contributions $I(s_i)$ while
> - (2) minimizing pairwise redundancy $\sum_{s_i, s_j \in S'} R(s_i, s_j)$.
>
> **PSCA approximates maximizing semantic importance (the First Term of Eq. (1))**
>
> PCA identifies the dominant variance (semantic) directions in the embedding space. Tokens with large projections on these principal components contribute more to the global semantic span. Ranking tokens by PSCA scores therefore provides a tractable approximation to maximizing the first term of Eq. (1), enabling the selection of tokens with high semantic significance.
>
> **NMS controls redundancy (the Second Term of Eq. (1))**
>
> Even among high-PSCA tokens, spatially adjacent or visually similar features may share large semantic overlap. NMS explicitly suppresses pairs whose cosine similarity exceeds a threshold $\epsilon$:
> $$
> R(s_i, s_j)\le \epsilon, \forall s_i,s_j\in S'.
> $$
> This ensures that retained tokens cover diverse semantic regions, directly reducing the second term of Eq. (1).
>
> ### (3) Joint Optimization: Why PSCA + NMS Achieve a Near-Optimal Balance
>
> Combining PSCA and NMS produces the following constrained objective:
>
> $$
> \text{Prune}\_{\text{ours}}(S) = \arg\max_{S' \subseteq S, |S'| = N}\sum_{s_i \in S'} I(s_i) \quad \text{subject to} \quad \forall_{i,j \in G_k}, R(s_i, s_j) \le \epsilon
> $$
>
> The resulting effective information is lower-bounded by:
> $$
> \text{Inform}(\text{Prune}\_{\text{ours}}(S)) \ge \max_{S' \subseteq S, |S'| = N} \left(\sum_{s_i \in S'} I(s_i)\right) - \sum_{k}\binom{|\tilde{G_k}|}{2} \cdot \epsilon,
> $$
> showing that redundancy is upper-bounded by a constant determined by $\epsilon$.
> Therefore, PSCA and NMS act as **complementary components** of a theoretically grounded pruning framework: PSCA promotes semantic importance, while NMS enforces diversity via redundancy control.
>
> ---
>
>
> We have incorporated this complete theoretical analysis into Sec. 3.5 of the main paper, and an extended version with full mathematical justification is included in Appendix 1.1.1 of the supplementary material. We hope this clarification fully addresses the reviewer’s concerns and makes the motivation behind PSCA–NMS much clearer and easier to understand.
>
> ## Q2. Qualitative and Failure Case Analysis
> Thank you for encouraging us to further analyze the qualitative behavior. We have added new visualizations：
> - **Fauliure cases**: in **Sec. 4.5 Visualization about Limitation** of the main paper, illustrating cases where PRUNESID may fail to preserve certain fine-grained visual attributes,
> - **Successful cases**: in **Appendix A.4.1: Visualization of PSCA Grouping and Retained Tokens**, showing examples where the method effectively retains semantically important tokens.
>
> (to be continue.)

---

> ### Author Response · Authors · 2025-11-24
> **To Reviewer t11S (Countinued)**
>
> ## Q3. Would Attention-Based or Saliency-Based Weighting Improve Interpretability?
> We appreciate the reviewer’s suggestion. Below we clarify 1) why attention-based weighting is not suitable for our setting, and 2) how our PSCA-based saliency mechanism already provides an interpretable and effective token-importance measure.
>
> - (1) Why **attention-based weighting** does not improve interpretability in our pruning setting?
>
>     Attention weights give a **global importance** signal and cannot directly support the **group-wise** pruning required by our method.
> - (2) How **PSCA-based saliency weighting** already provides an interpretable and pruning-compatible importance measure?
>
>     PSCA decomposes tokens into semantically coherent **groups** and assigns each token a **saliency weight score** reflecting its contribution within that semantic component. These scores are used directly in the NMS stage to keep the most informative token per group while removing redundant ones. This group-wise saliency formulation naturally enhances interpretability and aligns with our redundancy-aware pruning objective.

---

> ### Author Response · Authors · 2025-11-28
>
> Dear Reviewer t11S,
> Thank you again for reviewing our work. With the rebuttal deadline approaching, we just wanted to check whether any additional clarification is needed on our side. We are happy to provide further details if needed.

---

### Public Comment · ~Ruiguang_Pei1 · 2025-11-24
**Discussion on the Core Contributions of the Paper**

Dear respected author,

Greetings.

It has been a great pleasure to study your work, from which we have benefited immensely. We would like to take this opportunity to share this paper, which was publicly released on arXiv in June of this year: **GreedyPrune**[1], and we sincerely hope to engage in a discussion with you regarding this research. Thank you.

In this paper, it formulate token pruning in Large Vision-Language Models (LVLM) as a combinatorial optimization problem that closely resembles the Maximum Weight Independent Set problem in form, and solve it heuristically using a greedy approach. Briefly, this method first determines the semantic salience of each visual token by measuring the cosine similarity between the last text token and the image tokens. Then, it iteratively and greedily select the most salient visual token in each step while removing tokens that are semantically similar to it from the candidate set, **which is also Non-Maximum Suppression (NMS).**

From our perspective, the key contribution declared in your work—solving the grouping problem via NMS—appears to share considerable similarities with approach of GreedyPrune, and we would appreciate the opportunity to discuss this further. In our understanding, the main difference lies in the first step: your method uses PSCA to distinguish different visual groups (where tokens within the same group are semantically similar), and then applies NMS within each group before performing inter-group aggregation. In contrast, GreedyPrune applies NMS directly across the entire image. **Thus, the distinction seems to lie primarily in whether grouping is applied or not.** We would be grateful if you could elaborate on the methodological differences between the two approaches.

Additionally, we would like to request that you conduct an **ablation study** to clarify **whether the grouping operation (the first step) or the NMS operation (the second step) contributes more significantly to the performance**. This would directly impact the articulation of the paper's core argument.

Finally, we would like to note that several recent studies, such as BTP[2], HOLOV[3], and CDPrune[4], have also incorporated both diversity and importance considerations into their pruning frameworks. These works currently represent the strongest baselines in this line of research. We would appreciate it if you could also share your insights regarding these related approaches.
Thank you once again, and we look forward to your valuable feedback.

- [1]GreedyPrune: Retenting Critical Visual Token Set for Large Vision Language Models (Arxiv25.06)
- [2]Balanced Token Pruning: Accelerating Vision Language Models Beyond Local Optimization(BTP, NeurIPS25)
- [3]Don't Just Chase "Highlighted Tokens" in MLLMs: Revisiting Visual Holistic Context Retention(HoloV, NeurIPS25)
- [4]Beyond Attention or Similarity: Maximizing Conditional Diversity for Token Pruning in MLLMs(CDPrune, NeurIPS25)

---

> ### Author Response · Authors · 2025-12-02
> **To Ruiguang Pei**
>
> We sincerely thank the commenter for the thoughtful discussion and for bringing GreedyPrune and other recent pruning approaches to our attention. We are pleased to clarify the conceptual and methodological distinctions and to summarize the corresponding empirical evidence in our paper.
>
> ---
>
> ## **1. GreedyPrune vs. Our Method**
>
> Although both GreedyPrune and our approach incorporate *NMS-style redundancy suppression*, the underlying mechanisms differ substantially in both **efficiency** and **design philosophy**.
>
> ### **(a) PSCA-based semantic grouping enables GPU-parallel NMS**
>
> A key innovation of our method is the introduction of **PSCA-based semantic grouping**, which partitions tokens into coarse semantic groups before applying NMS. This design has two practical advantages:
>
> - **Inter-group parallelism**:
>   Because NMS is executed *independently within each group*, all groups can be processed in parallel on GPU. This contrasts with GreedyPrune, where global NMS is inherently sequential.
>
> - **Significantly reduced NMS iterations**:
>   As shown in *Table 16 (Appendix)*, each PSCA group retains only ~1–5 tokens after selection, so the number of NMS iterations is extremely small.
>   In GreedyPrune, global NMS operates over the full token set, typically requiring **tens to hundreds** of greedy iterations, making it computationally expensive.
>
> In short, while GreedyPrune performs global combinatorial search, our grouping–then–intra-group NMS design enables **highly efficient, batched GPU execution**, which is crucial for real-time VLM inference.
>
> ### **(b) Conceptual difference**
>
> - **GreedyPrune:** treats token pruning as a maximum-weight independent set problem and performs *global greedy selection + global suppression*.
> - **Ours:** separates the problem into:
>   1. **Semantic structure discovery** via PSCA;
>   2. **Local redundancy suppression** via intra-group NMS;
>   3. **Cross-group aggregation** for final token set.
>
> This two-stage decomposition not only improves efficiency but also yields more consistent retention of semantically diverse visual evidence.
>
> ---
>
> ## **2. Ablation on PSCA and NMS**
>
> We appreciate the request for ablations. As shown in *Table 13 (Appendix)*, we have already provided a direct comparison of removing PSCA and removing NMS:
>
> - **Removing PSCA** results in a significantly larger performance drop than removing NMS.
> - This indicates that **semantic grouping contributes more strongly** to accuracy retention than redundancy suppression alone.
>
> This finding also aligns with the intuition that importance and diversity must be balanced, and PSCA provides the structural prior necessary to achieve this balance effectively.
>
> ---
>
> ## **3. Regarding Recent Studies (BTP, HOLOV, CDPrune)**
>
> These works were released only recently, and due to the production timeline of our submission, we were unable to conduct full experimental reproduction or benchmarking. However, in the revision we have expanded the **Related Work** section to more clearly categorize these methods.
>
> We appreciate the commenter for highlighting these studies and will consider incorporating more comprehensive comparisons in future iterations of this research.

---

### Author Response · Authors · 2025-12-02
**Summary of rebuttal and revisions**

Dear Area Chair and Reviewers,

We sincerely thank all reviewers and the area chair for your considerable efforts.
Due to the unexpected circumstances, we provide here a concise summary of our rebuttal and revisions to help the area chair quickly grasp the key updates.

---

## 1. Consensus on Strengths
Across all reviewers, the following strengths were consistently highlighted:
- **Creative and practical method** (R-t11S, R-tbzB, R-q4vZ).
    The combination of PSCA and intra-group NMS provides a clean, interpretable, and training-free framework that is easy to integrate into existing VLMs.
- **Strong empirical performance with wide applicability** (R-t11S, R-ubst, R-q4vZ).
     The method is validated across diverse model families (LLaVA-1.5/NeXT, Mini-Gemini, Qwen2-VL/2.5-VL), parameter scales (7B/13B), and tasks (image/video), demonstrating strong generalization capability.
- **Thorough experimental study and ablations** (R-t11S, R-ubst, R-q4vZ).
    Extensive benchmark evaluations and detailed ablation studies convincingly support the method’s effectiveness.
- **Clear presentation and intuitive design** (R-t11S, R-ubst, R-tbzB, R-q4vZ).
     Reviewers praised the clear pipeline structure and effective visualization of semantic grouping and pruning.

---

## 2. Addressed Weaknesses and Revisions
Below we summarize the major concerns and questions from reviewers and how we addressed them.
### (1) Limited Theoretical Grounding of PSCA+NMS (R-t11S)
- Add a complete theoretical derivation based on the Inclusion–Exclusion Principle.
- Clarify how PSCA approximates maximizing semantic information and NMS explicitly minimizes redundancy.
- Expand Sec. 3.5 and add a full version in Appendix A.1.1.

### (2) Need qualitative and failure-case visualizations (R-t11S)
- Provide failure cases visualization and analysis in Sec. 4.5.
- Provide successful PSCA grouping and retained-token visualizations in Appendix A.4.1.

### (3) Request results on Qwen2.5-VL using high-resolution and more challenging benchmarks.  (R-ubst, R-q4vZ)
- Provide comprehensive results on Qwen2.5-VL 7B, including high-resolution benchmarks HRBench-8K and XLRS-macro, and more challenging benchmarks MMStar and MathVista.
- PRUNESID consistently outperforms VisionZip under all retention budgets.
- Provide experiment results in Appendix A.2.3.

### (4) Insufficient Baselines and Unverified Cross-Model Generalization (R-tbzB)
- Expand the baseline comparisons, and PRUNESID remains competitive or superior across 64/128/192-token budgets.
- Clarify the broad generalization experiments across LLaVA-1.5/NeXT (7B & 13B), Mini-Gemini (7B), and Qwen2-VL / Qwen2.5-VL has been provided **in appendix**.

### (5) Interaction with PDrop & Layer-wise Redundancy (R-ubst)
- Add new experiments showing that PRUNESID + PyramidDrop significantly outperforms PRUNESID-only under extremely tight token budgets (8–24 tokens).
- Demonstrate strong complementarity between V-encoder-stage and LLM-stage compression.
- Confirm that PDrop’s layer-wise redundancy phenomenon persists after PRUNESID compression.

---

## Final Remarks

We have carefully addressed all reviewer comments through new theoretical analysis, additional experiments, expanded baselines, and clearer visualizations. These revisions make the paper stronger and easier to understand. All updated or newly added content is marked in blue in the revised manuscript.

---

### Meta-Review · Area_Chair_LJjp · 2026-01-05

**Summary:**

After reading the paper and the rebuttal, most concerns are addressed by the authors. The AC tends to accept this paper.

**Reviewer Concerns:**

Most concerns are addressed.

**Reviewer Scores:**

Three of the reviewers gave a positive initial score.
For reviewer q4vZ, the concern about more benchmarks and methods is solved by sufficient results in the rebuttal. For other concerns, the authors also replied properly. So the score may rise.

---

### Decision · Program_Chairs · 2026-01-26

Accept (Poster)